# Stabilized E(n)-Equivariant Graph Neural Networks-assisted Generative Models

## Abstract

Due to its simplicity and computational efficiency, the E(n)-equivariant graph neural network (EGNN) [Satorras, et al., ICML, 2021] has been used as the backbone of equivariant normalizing flows (ENF), equivariant diffusion model (EDM), and beyond for Euclidean equivariant generative modeling. Nonetheless, it has been observed that ENF and EDM can be unstable; in this paper, we investigate the source of their instability by performing a sensitivity analysis of their backpropagation. Based on our theoretical analysis, we propose a regularization to stabilize and improve ENF and EDM. Experiments on benchmark datasets demonstrate that the regularized ENF outperforms the baseline model in terms of stability and computational efficiency by a remarkable margin. Furthermore, our results show that the proposed regularization can stabilize EDM and improve its performance.

## 1 Introduction

Graph neural networks (GNNs) have achieved remarkable success in various areas of molecular science, including molecular property prediction [14; 6; 4], molecular dynamics simulation [2; 30], and molecule generation [20; 40; 11; 39; 27; 28]. The core of GNNs are twofold: (1) to represent the input using a graph, e.g., encode a molecule as a graph with nodes and edges representing atoms and chemical bonds, respectively, and (2) to harness message passing on graphs for feature learning. Another crucial aspect is the preservation of 3D Euclidean symmetries such as translations, rotations, and reflection, collectively referred to as the 3D Euclidean group E(3).

Existing symmetry-aware GNNs under E(3) can be categorized based on the complexity of feature propagation. Invariant models like SchNet [37], DimeNet [10], and SphereNet [23] propagate features that remain invariant to E(3) transformations. Alternatively, equivariant models, e.g., the E(n)-equivariant GNN (EGNN) [36], update features that are equivariant to E(3) transformations during message passing. This equips them with the capability to capture the molecular dynamics under E(3) transformations. Additionally, steerable models [41; 8; 7; 22; 3] are a subset of equivariant models tailored to learn geometric and physical tensors. Among these symmetry-aware models, ***EGNN stands out due to its computational efficiency and capability to capture both molecular properties and dynamics***, making EGNN appealing for molecular generative modeling; see, e.g., ENF [35], EDM [18], GeoDiff [45], SMCDiff [42], and GEOLDM [46].

### 1.1 A recap on EGNN

Given a graph $\mathcal{G} = (\mathcal{V}, \mathcal{E})$ with $m$ nodes $v_i \in \mathcal{V}$ for $i = 1, \ldots, m$ and edges $e_{ij} \in \mathcal{E}$. Let $\boldsymbol{h}_i^0 \in \mathbb{R}^d$, $\boldsymbol{x}_i^0 \in \mathbb{R}^n$ be the input feature and spatial coordinate of $v_i$, respectively. The authors of [36] propose EGNN, which stacks the following E(n)-equivariant graph convolutional layers (EGCLs):

$$
\begin{aligned}
\boldsymbol{m}_{ij}^l &= \phi_e(\boldsymbol{h}_i^l, \boldsymbol{h}_j^l, \|\boldsymbol{x}_i^l - \boldsymbol{x}_j^l\|^2, a_{ij}), \\
\boldsymbol{x}_i^{l+1} &= \boldsymbol{x}_i^l + \sum_{j \in \mathcal{N}(i)} \frac{(\boldsymbol{x}_i^l - \boldsymbol{x}_j^l)}{\|\boldsymbol{x}_i^l - \boldsymbol{x}_j^l\| + 1} \phi_x(\boldsymbol{m}_{ij}^l), \\
\boldsymbol{m}_i^l &= \sum_{j \in \mathcal{N}(i)} \boldsymbol{m}_{ij}^l, \\
\boldsymbol{h}_i^{l+1} &= \phi_h(\boldsymbol{h}_i^l, \boldsymbol{m}_i^l),
\end{aligned}
\tag{1}
$$

where $\boldsymbol{x}_i^l, \boldsymbol{h}_i^l$ are the coordinate and features of $v_i$ at the $l^{th}$ layer, $\mathcal{N}(i) := \{v_j \in V \mid (v_i, v_j) \in \mathcal{E}\}$ is the set of neighbors of the $i^{th}$ node, $a_{ij}$ represents edge attributes, and $\phi_e, \phi_x, \phi_h$

are multi-layer perceptions (MLPs). The EGCL in (1) is adopted from [35; 18], which uses $\sum_{j \in \mathcal{N}(i)} \frac{(\boldsymbol{x}_i^l - \boldsymbol{x}_j^l)}{\|\boldsymbol{x}_i^l - \boldsymbol{x}_j^l\| + 1} \phi_x(\boldsymbol{m}_{ij}^l)$ rather than $\sum_{j \in \mathcal{N}(i)} (\boldsymbol{x}_i^l - \boldsymbol{x}_j^l) \phi_x(\boldsymbol{m}_{ij}^l)$ for the coordinate update. The normalized coordinate update can avoid drastic changes in coordinates during the generative process, and thus mitigate instability of the forward propagation. [36] has shown that EGNN—comprised of $L$ EGCLs—is E(n)-equivariant w.r.t. the coordinates $\boldsymbol{x} = (\boldsymbol{x}_1, \ldots, \boldsymbol{x}_m)$ while E(n)-invariant w.r.t. the features $\boldsymbol{h} = (\boldsymbol{h}_1, \ldots, \boldsymbol{h}_m)$, i.e., we have $\boldsymbol{R}\boldsymbol{x}^L, \boldsymbol{h}^L = \text{EGNN}(\boldsymbol{R}\boldsymbol{x}^0, \boldsymbol{h}^0)$, where $\boldsymbol{R}$ is an E(n) transformation, $\boldsymbol{R}\boldsymbol{x}^L := (\boldsymbol{R}\boldsymbol{x}_1^L, \ldots, \boldsymbol{R}\boldsymbol{x}_m^L)$, and $\boldsymbol{h}^L := (\boldsymbol{h}_1^L, \ldots, \boldsymbol{h}_m^L)$.

## 1.2  EGNN FOR GENERATIVE MODELING AND ITS INSTABILITY ISSUE

In this subsection, we review E(n)-equivariant normalizing flows (ENF) [35] and E(3)-equivariant diffusion models (EDM) [18]—two EGNN-assisted generative models. We then illustrate their training instability, which comes from the backpropagation of EGNNs—a fundamental issue in general EGNN-assisted generative models. Despite the instability issue, we discuss the advantages of using EGNN as the backbone of ENF and EDM over other symmetry-aware models in Section 2.

**ENF.** The generative process of ENF is defined by an invertible transformation $g(\boldsymbol{z}_h, \boldsymbol{z}_x) = (\boldsymbol{h}, \boldsymbol{x})$, mapping from a simple distribution $p_z(\boldsymbol{z}_h, \boldsymbol{z}_x)$ defined over latent features $\boldsymbol{z}_h \in \mathbb{R}^d$ and latent coordinates $\boldsymbol{z}_x \in \mathbb{R}^n$ to the data distribution $p(\boldsymbol{h}, \boldsymbol{x})$. To learn this invertible transformation, we treat $\boldsymbol{h}, \boldsymbol{x}$ as functions of time with $\boldsymbol{h}(t = 0) = \boldsymbol{h}$ and $\boldsymbol{x}(t = 0) = \boldsymbol{x}$. Then we model their dynamics towards latent representations, where $\boldsymbol{h}(1) = \boldsymbol{z}_h$ and $\boldsymbol{x}(1) = \boldsymbol{z}_x$, by the following ODE:

$$\frac{d\boldsymbol{h}(t)}{dt}, \frac{d\boldsymbol{x}(t)}{dt} = \boldsymbol{h}^L(t), \boldsymbol{x}^L(t) - \boldsymbol{x}(t), \tag{2}$$

where $\boldsymbol{h}(t), \boldsymbol{x}(t)$ are the input to a $L$-layer EGNN model and $\boldsymbol{h}^L(t), \boldsymbol{x}^L(t)$ are the output. Moreover, we have the following continuous-time change of variables formula for the log-likelihood:

$$\log p_v(\boldsymbol{h}, \boldsymbol{x}) = \log p_z(\boldsymbol{z}_h, \boldsymbol{z}_x) + \int_0^1 \text{Tr}\boldsymbol{J}_\phi\big(\boldsymbol{h}(t), \boldsymbol{x}(t)\big)dt, \tag{3}$$

where $\text{Tr}\boldsymbol{J}_\phi\big(\boldsymbol{h}(t), \boldsymbol{x}(t)\big)$ denotes the trace of the Jacobian matrix $\boldsymbol{J}_\phi\big(\boldsymbol{h}(t), \boldsymbol{x}(t)\big)$.

Notice that ENF directly considers the output $\boldsymbol{h}^L$ from the last layer of the EGNN as the derivative $\frac{d}{dt}\boldsymbol{h}(t)$ of the node features since the representation is invariant. In contrast, the derivative of the node coordinates w.r.t. time is computed as the difference between the EGNN's output and input, which guarantees the E(n)-equivariance of the coordinates.

**EDM.** EDM first defines a diffusion process to add noise to both node coordinates and features following a schedule such that both positions and features become Gaussian noise eventually. Then EDM learns a denoising process using a $L$-layer EGNN model to recover coordinates and features from the Gaussian noise. After the model is trained, the denoising EGNN can be used for molecule generation with Gaussian noise input. The technical details of EDM can be found in [18].

**Instability of ENF and EDM.** Despite using the normalized coordinate update as in (1), we observe that the instability of ENF and EDM remains. In particular, the gradient can change abruptly during the training. Let $\mathcal{L}$ be the loss function used for training ENF or EDM. Figure 1 plots the norm of the gradient $\|\partial\mathcal{L}/\partial\theta\|$ per training

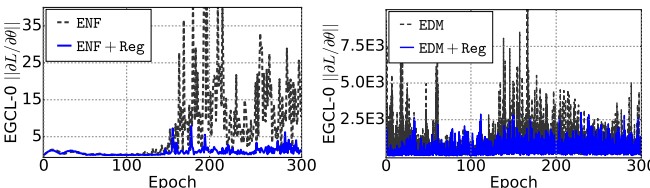

Figure 1: Norm of the gradient update of the input EGCL for ENF (left) and EDM (right), with and without regularization, during training on DW4 with 100 training samples.

iteration for ENF and EDM with $\theta$ being the parameters in the first layer of EGNN. This task was performed on the DW4 dataset with 100 training samples described in Sections 5.1 and 5.2 for ENF and EDM, respectively. Figure 1 shows that the gradient norm becomes very oscillatory as the training proceeds, making training unstable. We discuss more details in Section 5.

## 1.3  OUR CONTRIBUTION

In both ENF and EDM, the graph node coordinates keep changing during the generative process. Existing approaches that use the normalized coordinate update can effectively stabilize the forward propagation by avoiding drastic coordinate changes due to large differences $\boldsymbol{x}_i^l - \boldsymbol{x}_j^l$ for all $j \in \mathcal{N}(i)$.

However, ***there is no guarantee that this normalization stabilizes the backpropagation***. As such, we investigate the source of the instability of ENF and EDM by performing a sensitivity analysis, focusing on how backpropagation responds to changes in $\|\boldsymbol{x}_i^l - \boldsymbol{x}_j^l\|$; see Section 3 for details. From this, we propose the following regularized loss $\mathcal{L}_R$ to stabilize and improve training ENF and EDM:

$$\mathcal{L}_R := \mathcal{L} + \lambda \sqrt{\sum_{i,j \in \mathcal{N}(i),l} \left( \frac{\partial \phi_x(\boldsymbol{m}_{ij}^l)}{\partial \|\boldsymbol{x}_i^l - \boldsymbol{x}_j^l\|^2} \right)^2}, \quad (4)$$

where $\mathcal{L}$ denotes the original loss for training ENF or EDM, the second term is the proposed regularization, and $\lambda > 0$ is a hyperparameter (details in Section 4). Minimizing the regularization loss $\mathcal{L}_R$ helps stabilize the training of ENF and EDM. Figure 1 shows that ENF+Reg and EDM+Reg (+Reg denotes the model with our proposed regularization) are much more stable in $\|\partial\mathcal{L}/\partial\theta\|$ compared to the baseline models, stabilizing model training; see Section 5 for experimental results.

Furthermore, the regularization brings additional benefits: For ENF, it helps reduce the Lipschitz constant of the forcing function within the ODEs, thereby accelerating the training process. In the case of EDM, it enables stable learning even using EGNN with unnormalized coordinate updates, making EDM more flexible with improved performance for generative modeling.

### 1.4 ADDITIONAL RELATED WORKS

To the best of our knowledge, our work is the first study of stabilizing and improving the performance of ENF and EDM. Besides the most related work on proposing EGNN [36] and its applications in equivariant diffusion models [35; 18; 45; 42; 46]. A broader line of related works is developing generative models for molecule generation. In particular, E(n)-equivariant layers have been integrated into the autoregressive generative models; see e.g. [25; 13]. Variational autoencoder (VAE)-based models have also been proposed to generate 3D atomic coordinates, e.g., the papers [26; 32] propose to directly generate 3D atomic coordinates but they are not E(3)-equivariant. Conditional VAEs [38], Wasserstein GANs [17], and normalizing flows [29] have also been used for the molecular generation. Many other models have been proposed for molecule generation; see e.g. [12; 11; 44; 38; 22; 9; 15].

### 1.5 ORGANIZATION

We organize the rest of this paper as follows: We discuss some rationale of using EGNN for generative modeling in Section 2. We analyze the instability of ENF and EDM in Section 3 and discuss the proposed regularization in Section 4. We verify the efficacy of our proposed regularization on a few benchmark tasks in Section 5. Technical proofs and additional details are provided in the appendix.

## 2 WHY DO WE USE EGNN FOR GENERATIVE MODELING?

Since the development of ENF and EDM, both models have gained remarkable attention. In this section, we discuss some advantages of using EGNN with the normalized coordinate update for ENF and EDM; the merits of both models have been thoroughly discussed in [35; 18]. To the best of our knowledge, EGNN is the most popular architecture to ensure equivariance in the context of Euclidean equivariant normalizing flows and diffusion models [35; 18; 45; 42; 46]. The remarkable success and impact of EGNN—especially in the context of generative modeling—makes it important to stabilize their training and improve it for better generative modeling.

### 2.1 EGNN VS. OTHER SYMMETRY-AWARE MODELS

There are three main classes of symmetry-aware GNNs under E(3): invariant models, equivariant models that update only scalars and vectors, and steerable models that update tensor features in message-passing. Firstly, invariant models do not learn positional features, and are therefore not suitable for generative modeling since coordinates update is crucial in molecular generation. Additionally, molecular generation inherently exhibits equivariance. Moreover, it has been shown that most invariant GNNs have lower expressive power than equivariant GNNs [21]. Secondly, generative models, such as normalizing flows and diffusion models, often entail an excessive number of evaluations of the neural network function and gradient computations. This prohibitive computational demand makes steerable models less practical for generative modeling. Indeed, the excessive computational expense of steerable models has been discussed in existing papers; see e.g. [3; 31].

The simplicity, expressivity, and computational efficiency of EGNN make it appealing for serving as the backbone of Euclidean equivariant generative models, resulting in ENF, EDM, and beyond. Besides EGNN, applying other existing computationally efficient or designing new equivariant GNNs for generative modeling can be an interesting future work.

## 2.2 Normalized vs. unnormalized coordinate updates in ENF and EDM

This subsection shows that using the normalized coordinate update can improve ENF and EDM over using the unnormalized one. Figure 2 compares ENF and EDM using the normalized and the unnormalized coordinate updates for the same task as in Figure 1. It shows that using the normalized coordinate update avoids the blowup problem that occurs in the unnormalized one.

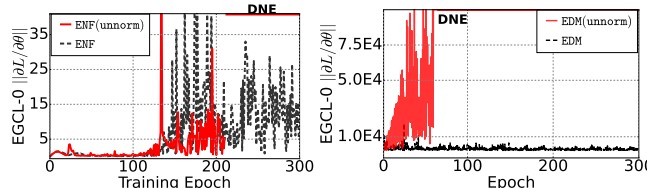

Figure 2: Norm of the gradient update of the input EGCL for ENF (left) and EDM (right), using normalized or unnormalized coordinate update, during training on DW4 with 100 training samples. DNE is the point at which the model blows up.

When using the unnormalized coordinate, all differences $\boldsymbol{x}_i^l - \boldsymbol{x}_j^l$ for all $j \in \mathcal{N}(i)$ are used to get the updated coordinate $\boldsymbol{x}_i^{l+1}$, which can result in drastic coordinate changes. The drastic coordinate change will be further propagated during forward propagation, resulting in a potential blowup. In contrast, normalizing coordinates can avoid abnormal coordinate updates from large differences among coordinates of neighboring nodes. *Nevertheless, the instability issue still occurs in backpropagation using the normalized coordinate update, which is crucial to be addressed.*

## 3 A Theoretical Study of the Instability Issues

In this section, we study the instability issue of training ENF and EDM by analyzing their backpropagation. In particular, we consider EGNN with $L$ EGCLs in (1) or using the following unnormalized coordinate updates:

$$\boldsymbol{x}_i^{l+1} = \boldsymbol{x}_i^l + \sum_{j \in \mathcal{N}(i)} (\boldsymbol{x}_i^l - \boldsymbol{x}_j^l)\phi_x(\boldsymbol{m}_{ij}^l). \tag{5}$$

The analysis in this section can be easily adapted to EGCL with the normalized coordinate update.

### 3.1 Backpropagation

We study the cause of instability of ENF and EDM in their backpropagation. As we will see, the instability issue not only occurs for ENF and EDM but also for other EGNN-assisted generative models since the instability comes from the EGNN. Let $\text{vec}(\cdot)$ denote the vectorization of the input matrix. Consider $\boldsymbol{h}^l := \text{vec}\left([\boldsymbol{h}_1^l, \boldsymbol{h}_2^l, \ldots, \boldsymbol{h}_m^l]\right)$, $\boldsymbol{x}^l := \text{vec}\left([\boldsymbol{x}_1^l, \boldsymbol{x}_2^l, \ldots, \boldsymbol{x}_m^l]\right)$, which consist of the node features and spatial coordinates, respectively. We define $\boldsymbol{f}^l := [(\boldsymbol{h}^l)^\top, (\boldsymbol{x}^l)^\top]^\top$ as the feature at the $l^{th}$ layer. Consider $\mathcal{L}(\boldsymbol{f}^L)$ to be the loss function or any function of $\boldsymbol{f}^L$ and let $\theta$ be any weight parameter appearing at the $l^{th}$ layer of EGNN. Applying the chain rule to the derivative $\frac{\partial \mathcal{L}(\boldsymbol{f}^L)}{\partial \theta}$, we see that

$$\frac{\partial \mathcal{L}(\boldsymbol{f}^L)}{\partial \theta} = \frac{\partial \mathcal{L}(\boldsymbol{f}^L)}{\partial \boldsymbol{f}^l}\frac{\partial \boldsymbol{f}^l}{\partial \theta} = \frac{\partial \mathcal{L}(\boldsymbol{f}^L)}{\partial \boldsymbol{f}^{l+1}}\frac{\partial \boldsymbol{f}^{l+1}}{\partial \boldsymbol{f}^l}\frac{\partial \boldsymbol{f}^l}{\partial \theta} = \frac{\partial \mathcal{L}(\boldsymbol{f}^L)}{\partial \boldsymbol{f}^L}\left(\prod_{l'=0}^{L-l-1} \frac{\partial \boldsymbol{f}^{L-l'}}{\partial \boldsymbol{f}^{L-l'-1}}\right)\frac{\partial \boldsymbol{f}^l}{\partial \theta}. \tag{6}$$

Notice that the derivative $\frac{\partial \mathcal{L}(\boldsymbol{f}^L)}{\partial \boldsymbol{f}^l}$ propagates backward through $\frac{\partial \boldsymbol{f}^{l+1}}{\partial \boldsymbol{f}^l}$, which motives us to investigate $\frac{\partial \boldsymbol{f}^{l+1}}{\partial \boldsymbol{f}^l}$; it suffices to examine $\frac{\partial \boldsymbol{h}_i^{l+1}}{\partial \boldsymbol{h}_j^l}$, $\frac{\partial \boldsymbol{x}_i^{l+1}}{\partial \boldsymbol{h}_j^l}$, $\frac{\partial \boldsymbol{h}_i^{l+1}}{\partial \boldsymbol{x}_j^l}$, and $\frac{\partial \boldsymbol{x}_i^{l+1}}{\partial \boldsymbol{x}_j^l}$ for any $1 \le i, j \le m$ since

$$\frac{\partial \boldsymbol{f}^{l+1}}{\partial \boldsymbol{f}^l} = \begin{bmatrix} \frac{\partial \boldsymbol{h}^{l+1}}{\partial \boldsymbol{h}^l} & \frac{\partial \boldsymbol{h}^{l+1}}{\partial \boldsymbol{x}^l} \\ \frac{\partial \boldsymbol{x}^{l+1}}{\partial \boldsymbol{h}^l} & \frac{\partial \boldsymbol{x}^{l+1}}{\partial \boldsymbol{x}^l} \end{bmatrix}.$$

Let $\boldsymbol{I}_m$ denote the identity matrix of size $m \times m$. In the following Proposition 1, we present the detailed form of $\frac{\partial \boldsymbol{h}_i^{l+1}}{\partial \boldsymbol{h}_j^l}$, $\frac{\partial \boldsymbol{x}_i^{l+1}}{\partial \boldsymbol{h}_j^l}$, $\frac{\partial \boldsymbol{h}_i^{l+1}}{\partial \boldsymbol{x}_j^l}$, and $\frac{\partial \boldsymbol{x}_i^{l+1}}{\partial \boldsymbol{x}_j^l}$ for EGCL with the unnormalized coordinate update.

**Proposition 1.** *Consider EGCL using the unnormalized coordinate update in (5), let* $\frac{\partial \phi_h(\boldsymbol{h}_i^l|\boldsymbol{m}_i^l)}{\partial \boldsymbol{h}_i^l} :=$ $\lim_{\boldsymbol{t} \to \boldsymbol{0}} \frac{\phi_h(\boldsymbol{h}_i^l+\boldsymbol{t},\boldsymbol{m}_i^l)-\phi_h(\boldsymbol{h}_i^l,\boldsymbol{m}_i^l)}{\boldsymbol{t}}$ *be the derivative of* $\phi_h(\boldsymbol{h}_i^l, \boldsymbol{m}_i^l)$ *w.r.t. the first input* $\boldsymbol{h}_i^l$. *Similarly, let* $\frac{\partial \phi_h(\boldsymbol{m}_i^l|\boldsymbol{h}_i^l)}{\partial \boldsymbol{m}_i^l} := \lim_{\boldsymbol{t} \to \boldsymbol{0}} \frac{\phi_h(\boldsymbol{h}_i^l,\boldsymbol{m}_i^l+\boldsymbol{t})-\phi_h(\boldsymbol{h}_i^l,\boldsymbol{m}_i^l)}{\boldsymbol{t}}$ *be the derivative of* $\phi_h(\boldsymbol{h}_i^l, \boldsymbol{m}_i^l)$ *w.r.t. the second*

input $\boldsymbol{m}_i^l$. Then we have

$$\frac{\partial \boldsymbol{h}_i^{l+1}}{\partial \boldsymbol{h}_j^l} = \begin{cases} \frac{\partial \phi_h(\boldsymbol{h}_i^l | \boldsymbol{m}_i^l)}{\partial \boldsymbol{h}_i^l} + \frac{\partial \phi_h(\boldsymbol{m}_i^l | \boldsymbol{h}_i^l)}{\partial \boldsymbol{m}_i^l} \sum_{k \in \mathcal{N}(i)} \frac{\partial \boldsymbol{m}_{ik}^l}{\partial \boldsymbol{h}_i^l} & \text{if } j = i, \\ \frac{\partial \phi_h(\boldsymbol{m}_i^l | \boldsymbol{h}_i^l)}{\partial \boldsymbol{m}_i^l} \frac{\partial \boldsymbol{m}_{ij}^l}{\partial \boldsymbol{h}_j^l} & \text{if } j \neq i, \end{cases}$$

$$\frac{\partial \boldsymbol{h}_i^{l+1}}{\partial \boldsymbol{x}_j^l} = \begin{cases} \frac{\partial \phi_h(\boldsymbol{m}_i^l | \boldsymbol{h}_i^l)}{\partial \boldsymbol{m}_i^l} \sum_{k \in \mathcal{N}(i)} \frac{\partial \boldsymbol{m}_{ik}^l}{\partial \|\boldsymbol{x}_i^l - \boldsymbol{x}_k^l\|^2} \cdot (\boldsymbol{x}_i^l - \boldsymbol{x}_k^l)^\top & \text{if } j = i, \\ -\frac{\partial \phi_h(\boldsymbol{m}_i^l | \boldsymbol{h}_i^l)}{\partial \boldsymbol{m}_i^l} \frac{\partial \boldsymbol{m}_{ik}^l}{\partial \|\boldsymbol{x}_i^l - \boldsymbol{x}_j^l\|^2} \cdot 2(\boldsymbol{x}_i^l - \boldsymbol{x}_j^l)^\top & \text{if } j \neq i, \end{cases}$$

$$\frac{\partial \boldsymbol{x}_i^{l+1}}{\partial \boldsymbol{h}_j^l} = \begin{cases} \sum_{k \in \mathcal{N}(i)} (\boldsymbol{x}_i^l - \boldsymbol{x}_k^l) \frac{\partial \phi_x(\boldsymbol{m}_{ik}^l)}{\partial \boldsymbol{m}_{ik}^l} \frac{\partial \boldsymbol{m}_{ik}^l}{\partial \boldsymbol{h}_i^l} & \text{if } j = i, \\ (\boldsymbol{x}_i^l - \boldsymbol{x}_j^l) \frac{\partial \phi_x(\boldsymbol{m}_{ij}^l)}{\partial \boldsymbol{m}_{ij}^l} \frac{\partial \boldsymbol{m}_{ij}^l}{\partial \boldsymbol{h}_j^l} & \text{if } j \neq i, \end{cases}$$

$$\frac{\partial \boldsymbol{x}_i^{l+1}}{\partial \boldsymbol{x}_j^l} = \begin{cases} \boldsymbol{I}_3 + \sum_{k \in \mathcal{N}(i)} \boldsymbol{G}_{ik}^l & \text{if } j = i, \\ -\boldsymbol{G}_{ij}^l & \text{if } j \neq i, \end{cases}$$

where $\boldsymbol{G}_{ij}^l := \phi_x(\boldsymbol{m}_{ij}^l) \boldsymbol{I}_3 + \frac{\partial \phi_x(\boldsymbol{m}_{ij}^l)}{\partial \|\boldsymbol{x}_i^l - \boldsymbol{x}_j^l\|^2} \cdot 2(\boldsymbol{x}_i^l - \boldsymbol{x}_j^l)(\boldsymbol{x}_i^l - \boldsymbol{x}_j^l)^\top$.

**Remark 1.** *By considering $\phi_x / (\|\boldsymbol{x}_i^l - \boldsymbol{x}_j^l\| + 1)$ as $\phi_x$ in (5), similar studies can be carried out for the EGCL with the normalization scheme in (1).*

## 3.2 SENSITIVITY ANALYSIS

In this subsection, we investigate the sensitivity of the backpropagation—governed by the derivative $\frac{\partial \boldsymbol{f}^{l+1}}{\partial \boldsymbol{f}^l}$—in response to $\|\boldsymbol{x}_i^l - \boldsymbol{x}_j^l\|$. We aim to identify which part of $\frac{\partial \boldsymbol{f}^{l+1}}{\partial \boldsymbol{f}^l}$ is most sensitive to $\|\boldsymbol{x}_i^l - \boldsymbol{x}_j^l\|$. We examine the sensitivities of the four partial derivatives $\frac{\partial \boldsymbol{h}_i^{l+1}}{\partial \boldsymbol{h}_j^l}$, $\frac{\partial \boldsymbol{x}_i^{l+1}}{\partial \boldsymbol{h}_j^l}$, $\frac{\partial \boldsymbol{h}_i^{l+1}}{\partial \boldsymbol{x}_j^l}$, and $\frac{\partial \boldsymbol{x}_i^{l+1}}{\partial \boldsymbol{x}_j^l}$ in $\|\boldsymbol{x}_i^l - \boldsymbol{x}_j^l\|$, and we characterize the extent of their sensitivity by determining the highest degree of the norm $\|\boldsymbol{x}_i^l - \boldsymbol{x}_j^l\|$ that can appear in the norm of these partial derivatives. We do not consider the term $\|\boldsymbol{x}_i^l - \boldsymbol{x}_j^l\|^2$ that appears as the input to the function $\phi_e$ since one can use a bounded activation function, such as tanh, to mitigate its causes of EGNN's instability. Based on the explicit forms of the partial derivatives in Propositions 1, we summarize their sensitivity as follows:

- $\partial \boldsymbol{h}_i^{l+1} / \partial \boldsymbol{h}_j^l$ does not contain $\boldsymbol{x}_i^l - \boldsymbol{x}_j^l$. It is not sensitive to $\|\boldsymbol{x}_i^l - \boldsymbol{x}_j^l\|$.
- $\partial \boldsymbol{x}_i^{l+1} / \partial \boldsymbol{h}_j^l$ contains $(\boldsymbol{x}_i^l - \boldsymbol{x}_j^l)^\top$. It is sensitive to $\|\boldsymbol{x}_i^l - \boldsymbol{x}_j^l\|$ of degree 1.
- $\partial \boldsymbol{h}_i^{l+1} / \partial \boldsymbol{x}_j^l$ contains $(\boldsymbol{x}_i^l - \boldsymbol{x}_j^l)$. It is sensitive to $\|\boldsymbol{x}_i^l - \boldsymbol{x}_j^l\|$ of degree 1.
- $\partial \boldsymbol{x}_i^{l+1} / \partial \boldsymbol{x}_j^l$ contains $(\boldsymbol{x}_i^l - \boldsymbol{x}_j^l)(\boldsymbol{x}_i^l - \boldsymbol{x}_j^l)^\top$. It is sensitive to $\|\boldsymbol{x}_i^l - \boldsymbol{x}_j^l\|$ of degree 2.

We see that $\frac{\partial \boldsymbol{x}_i^{l+1}}{\partial \boldsymbol{x}_j^l}$ is the most sensitive term to $\|\boldsymbol{x}_i^l - \boldsymbol{x}_j^l\|$, and $(\boldsymbol{x}_i^l - \boldsymbol{x}_j^l)(\boldsymbol{x}_i^l - \boldsymbol{x}_j^l)^\top$ is multiplied by the scalar $\frac{2 \partial \phi_x(\boldsymbol{m}_{ij}^l)}{\partial \|\boldsymbol{x}_i^l - \boldsymbol{x}_j^l\|^2}$ in the explicit expression of $\frac{\partial \boldsymbol{x}_i^{l+1}}{\partial \boldsymbol{x}_j^l}$. Thus, $\frac{\partial \phi_x(\boldsymbol{m}_{ij}^l)}{\partial \|\boldsymbol{x}_i^l - \boldsymbol{x}_j^l\|^2}$ *is directly related to the sensitivity of* $\frac{\partial \boldsymbol{x}_i^{l+1}}{\partial \boldsymbol{x}_j^l}$ *and hence the sensitivity of the backpropagation w.r.t.* $\|\boldsymbol{x}_i^l - \boldsymbol{x}_j^l\|$.

## 4 REGULARIZED EGNN FOR GENERATIVE MODELING

Let $\mathcal{L}$ be the loss used for training ENF or EDM. We aim to stabilize the training process of ENF and EDM by controlling the sensitivity of the gradient $\frac{\partial \mathcal{L}(\boldsymbol{f}^L)}{\partial \theta}$ w.r.t. $\|\boldsymbol{x}_i^l - \boldsymbol{x}_j^l\|$, where $\theta$ denotes any learnable parameter of the model. As discussed in Section 3.2, regularizing the magnitude of $\frac{\partial \phi_x(\boldsymbol{m}_{ij}^l)}{\partial \|\boldsymbol{x}_i^l - \boldsymbol{x}_j^l\|^2}$ can help us achieve this goal. Therefore, we propose a regularization scheme for ENF and EDM by adding a penalty term to the loss function $\mathcal{L}$, resulting in the following regularized loss function $\mathcal{L}_R$ in (4). Notice that the penalty term is an L2 regularization on all scalars $\frac{\partial \phi_x(\boldsymbol{m}_{ij}^l)}{\partial \|\boldsymbol{x}_i^l - \boldsymbol{x}_j^l\|^2}$ for all $i, j, l$. Therefore, minimizing $\mathcal{L}_R$ will encourage $\frac{\partial \phi_x(\boldsymbol{m}_{ij}^l)}{\partial \|\boldsymbol{x}_i^l - \boldsymbol{x}_j^l\|^2}$ for all $i, j, l$ to be small. By using the regularized loss function in (4), we expect the training of both ENF and EDM will be more stable

than using the unregularized loss function. In the rest of this section, we discuss some additional benefits of using the proposed regularization.

### 4.1 OTHER BENEFITS OF REGULARIZATION TO ENF

In the following, we discuss the proposed regularization can accelerate training ENF, which requires solving ODEs numerically. Let $\phi$ be an ENF in (2) with the right-hand side (forcing function) modeled by a $L$-layer EGNN model. Again, let $\boldsymbol{f}^l := [(\boldsymbol{h}^l)^\top, (\boldsymbol{x}^l)^\top]^\top$ denotes the feature at the $l^{th}$ layer of EGNN and $\phi(\boldsymbol{f}^0) := \boldsymbol{f}^L - [0, (\boldsymbol{x}^0)^\top]^\top$. Notice that the stability and accuracy of numerical solutions of the ODE are related to the Lipschitz constant of the forcing function of (2). In particular, when using the benchmark and default DOPRI solver [5] or other adaptive stepsize solvers for solving the ODEs, the error estimation of the ODE solver is controlled by the (local) Lipschitz constant of the function, and such an estimated error will be used to determine the step size— a smaller Lipschitz constant allows using larger step sizes [16], reducing the number of function evaluations (NFEs) and improving computational efficiency. Moreover, the spectral norm of the Jacobian matrix $\|\boldsymbol{J}_\phi\|_2$ can be used to estimate the Lipschitz constant of $\phi$. Maintaining a small and stable spectral norm throughout the numerical solution of the ODE benefits stability, accuracy, and computational efficiency. Our regularization can stabilize $\|\boldsymbol{J}_\phi\|_2$; in particular, we observe that

$$\boldsymbol{J}_\phi = \frac{\partial \phi(\boldsymbol{f}^0)}{\partial \boldsymbol{f}^0} = \frac{\partial \boldsymbol{f}^L}{\partial \boldsymbol{f}^0} - \frac{\partial [0, (\boldsymbol{x}^0)^\top]^\top}{\partial \boldsymbol{f}^0} = \prod_{l=0}^{L} \frac{\partial \boldsymbol{f}^{l+1}}{\partial \boldsymbol{f}^l} - \begin{bmatrix} \boldsymbol{O} & \boldsymbol{O} \\ \boldsymbol{O} & \boldsymbol{I}_m \end{bmatrix} \tag{7}$$

where $\boldsymbol{O}$ denotes a matrix consisting of zeros. Since the regularized loss $\mathcal{L}_R$ is designed to stabilize each term $\partial \boldsymbol{f}^{l+1}/\partial \boldsymbol{f}^l$, we conclude that our regularization can be beneficial for solving the ODE in ENFs. It encourages a stable and well-behaved spectral norm $\|\boldsymbol{J}_\phi\|_2$, resulting in improved numerical stability, accuracy, and efficiency. We will numerically verify this in Section 5.1.

### 4.2 OTHER BENEFITS OF REGULARIZATION TO EDM

Proposition 1 shows that our proposed regularization can stabilize the backpropagation of EDM even using EGNN with the unnormalized coordinate update in (5). Moreover, we notice empirically that the regularization can also stabilize the forward propagation of EDM when using EGNN with the unnormalized coordinate update. In practice, using the unnormalized coordinate update makes the coordinate update more flexible in the generative process, which improves the performance of EDM; see experimental results in Section 5.2. However, from our experiments, we do not see that the regularization helps stabilize the forward propagation of ENF using the unnormalized coordinate update since the complicated numerical integration is involved, and small instability can be propagated and amplified.

## 5 EXPERIMENTS

In this section, we demonstrate that our regularization (1) improves the stability and computational efficiency of ENF and (2) stabilizes EDM with improved accuracy and does not significantly raise the computational cost. For normalizing flows, we compare ENF+Reg against ENF. For diffusion models, we compare the performance of EDM, EDM(unnorm) (replacing the normalized coordinate update with the unnormalized one), their regularized versions—denoted as EDM+Reg and EDM(unnorm)+Reg, and a few very recent equivariant diffusion models. Three benchmark tasks are used for comparison, namely, DW4 [22], LJ13 [22], and QM9 [34]. We train the model by minimizing NLL and the regularized loss function (4) for the baseline and regularized models, respectively, using the Adam optimizer. Experiments were run using NVIDIA RTX3090 and timing experiments were run using Google Colab [24] A100 GPU. We provide additional experimental details in Appendix B.

### 5.1 EQUIVARIANT NORMALIZING FLOWS

We demonstrate the benefits of regularization in improving the stability and computational efficiency of ENF. We select the optimal regularization hyperparameter $\lambda$ via grid search.

#### 5.1.1 DW4 AND LJ13

The DW4 and LJ13 datasets introduced in [22] sample the positions—using double-well and Lennard-Jones potential—of $n$ atoms where $n$ is 4 and 13 for DW4 and LJ13, respectively. We use the fixed data splits of $\{10^2, 10^3, 10^4\}$ training, 1000 validation, and 1000 testing data samples for both datasets. Following the training procedure in [35], we use Adam with learning rate $5e$-3, weight decay $1e$-12, and batch size 100. We utilize Hutchinson's trace estimator [19] in the training

procedure. In validation and testing, we use the exact trace to estimate the negative log-likelihood (NLL). The forcing function of ENF is selected to be an EGNN with 6 layers, 32 features per layer, and SiLU activation. The optimal regularization hyperparameter from the grid search is $\lambda = 0.02$.

**Stabilized training.** Figure 1 shows that the proposed regularization can avoid very large gradients. We further show that using the proposed regularization can effectively alleviate large spikes in training loss. As shown in Figure 3(a) and (b), compared to the training curves of ENF for DW4 and LJ13, the loss curves are smoother for ENF+Reg; in particular, the spikes are much larger in the loss curves for unregularized ENF compared to the regularized ones.

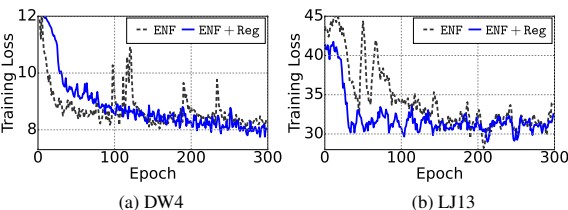

(a) DW4      (b) LJ13

Figure 3: Training loss for ENF and ENF+Reg on DW4 and LJ13 using $10^4$ training samples.

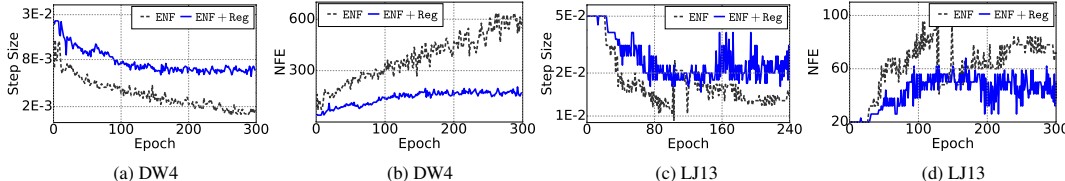

(a) DW4      (b) DW4      (c) LJ13      (d) LJ13

Figure 4: Comparing ENF+Reg against ENF in averaged step size and number of function evaluations (NFEs) per training epoch used by the ODE solver on DW4 (a,b) and LJ13 (c,d).

**Accelerated training.** Our analysis in Section 4.1 indicates that the regularization allows the use of larger step sizes for the ODE solver and reduces NFEs and computational time compared to training ENF without regularization. We verify the benefit of regularization using DW4 and LJ13 each with $10^4$ training samples. Figure 4 plots the step size and NFEs used by the ODE solver[1], confirming that regularization can remarkably improve computational efficiency. We report the average time per epoch in Table 1.

| Split | DW4 | | | LJ13 | | |
|---|---|---|---|---|---|---|
| | $10^2$ | $10^3$ | $10^4$ | $10^2$ | $10^3$ | $10^4$ |
| ENF | 23.8 | 73.9 | 910.7 | 37.0 | 54.0 | 554.1 |
| ENF+Reg | 8.6 | 38.2 | 401.7 | 24.9 | 39.5 | 332.9 |

Table 1: Average time per epoch for ENF and ENF+Reg for each split of DW4 and LJ13. Unit: second.

| Split | DW4 | | | LJ13 | | |
|---|---|---|---|---|---|---|
| | $10^2$ | $10^3$ | $10^4$ | $10^2$ | $10^3$ | $10^4$ |
| ENF | $12.28_{\pm 0.2}$ | $\mathbf{8.35_{\pm 0.2}}$ | $7.68_{\pm 0.1}$ | $31.52_{\pm 0.1}$ | $\mathbf{31.02_{\pm 0.2}}$ | $30.30_{\pm 0.3}$ |
| ENF+Reg | $\mathbf{12.02_{\pm 0.2}}$ | $8.47_{\pm 0.2}$ | $\mathbf{7.65_{\pm 0.1}}$ | $\mathbf{31.19_{\pm 0.2}}$ | $31.05_{\pm 0.2}$ | $\mathbf{29.66_{\pm 0.3}}$ |

Table 2: Negative log-likelihood for ENF and ENF+Reg for each split of DW4 and LJ13.

**Performance.** We compare the performance of ENF and its regularized version for DW4 and LJ13 generation. Table 2 reports the test NLL of ENF and ENF+Reg for different sizes of training data. We report the test NLL at the best validation NLL after the model is trained for 300 epochs. These results show that the regularization does not degrade the performance of ENF.

We further analyze the generative properties of ENF and ENF+Reg by contrasting the true and sampled molecules on DW4 and LJ13 following [35]. Figure 5 shows the probability distribution with mean and standard deviation of the relative distances for true and sampled molecules from LJ13 using ENF and ENF+Reg trained with 100 samples. We see that ENF+Reg results in a mean that is closer to the ground truth and the standard deviations are equivalent. We show similar results for DW4 and the molecular energies in Appendix B.

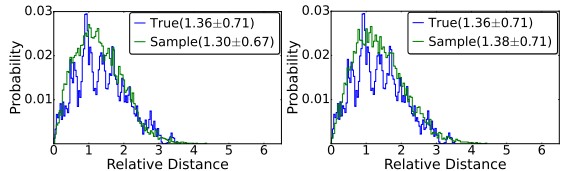

Figure 5: Probability distributions (mean±std dev) of the relative distance for 100 true and sampled molecules on LJ13 using ENF(left) and ENF+Reg(right) with 100 training samples.

---

[1]Here, for better visualization, we plot the averaged step size and NFE per epoch.

### 5.1.2 QM9

We consider QM9 [33]—a dataset contains molecular properties and atom coordinates for 130k small molecules with up to 9 heavy atoms (29 atoms including hydrogens). We train ENF and ENF+Reg to unconditionally generate molecules with 3D coordinates, atom types (H, C, N, O, F), and integer-valued atom charges. We use the train/validation/test partitions introduced in [1], consisting of 100K/18K/13K samples, respectively. We use Hutchinson's trace estimator [19] in training and use the estimated NLL in validation and testing. We set the learning rate as $2e$-4 and weight decay of $1e$-12 for Adam with batch size 64. For regularization, we set $\lambda = 1e$-2. We train both ENF and ENF+Reg for 300 epochs. We set the EGNN model using 6 layers, 256 features per layer, and SiLU activation.

We use the distance between pairs of atoms and the atom types to predict bond types (single, double, triple, or none). We measure atomic stability (the proportion of atoms that have the right valency) and molecular stability (the proportion of generated molecules for which all atoms are stable) [35]. Table 4 compares the performance of ENF and ENF+Reg, showing that regularization improves ENF by a noticeable margin in test NLL and atomic and molecular stability of the generated molecules. Meanwhile, regularization also accelerates learning the model.

### 5.2 Equivariant diffusion models

In this subsection, we aim to show that regularization can stabilize EDM and improve its performance. The regularized EDM requires some more computational cost than EDM but is more efficient compared to some most recent models. Moreover, we show that regularization allows EDM to use EGNN with the unnormalized coordinate update, resulting in better performance.

### 5.2.1 DW4 and LJ13

We train EDM, EDM(unnorm), EDM+Reg, and EDM(unnorm)+Reg with different numbers of training samples using Adam with learning rate $5e$-4, weight decay $1e$-12, and batch size 100 for 300 epochs, and using 1000 diffusion steps. In these experiments, the EDM contains 3 EGNN blocks and each block consists of 2 EGCL layers and 1 coordinates equivariant update step, 32 features per layer, and SiLU activation. The optimal regularization hyperparameters from the grid search are $1e$-4 and $5e$-4 for EDM+Reg and EDM(unnorm)+Reg, respectively.

**Stabilized training.** We show that regularization can stabilize training EDM even using EGNN with the unnormalized coordinate update. Figure 6 compares the training curves of the four models on both DW4 and LJ13 with $10^4$ training data. EDM(unnorm) blows up quickly as the training proceeds. All the other three models do not blow up and regularization makes the training curve smoother.

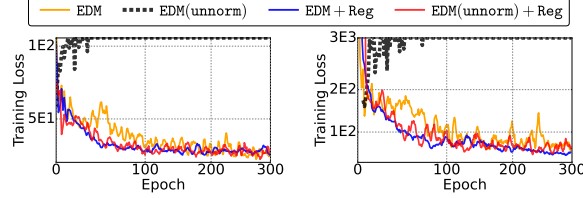

Figure 6: Training loss for EDM, EDM(Unnrom), EDM+reg and EFN(Unnrom)+Reg over epochs on DW4 (left) and LJ13 (right).

**Performance and computational cost.** Table 3 contrasts NLL and computational cost of the four models, showing that regularization can improve the performance of both EDM and EDM(unnorm) by a noticeable margin and does not significantly raise the computational cost. Moreover, except for LJ13 with $10^4$ training samples, EDM(unnorm)+Reg performs the best among all four models.

| | **DW4** | | | | **LJ13** | | | |
|---|---|---|---|---|---|---|---|---|
| **Metrics** Split | $10^3$ | | $10^4$ | | $10^3$ | | $10^4$ | |
| EDM(Unnorm) | $25.05_{\pm 1.9}$ | (1.2) | Nan | ( − ) | Nan | ( − ) | Nan | ( − ) |
| EDM | $22.73_{\pm 1.7}$ | (1.1) | $14.26_{\pm 0.6}$ | (11.3) | $46.77_{\pm 5.2}$ | (1.0) | $15.05_{\pm 1.5}$ | (8.8) |
| EDM(Unnorm)+Reg | $\mathbf{20.57_{\pm 2.3}}$ | (1.5) | $\mathbf{12.45_{\pm 0.8}}$ | (16.1) | $\mathbf{32.21_{\pm 4.1}}$ | (1.4) | $12.85_{\pm 1.3}$ | (13.6) |
| EDM+Reg | $22.39_{\pm 0.8}$ | (1.6) | $13.36_{\pm 1.1}$ | (15.3) | $39.13_{\pm 4.1}$ | (1.3) | $\mathbf{11.62_{\pm 1.8}}$ | (13.6) |

Table 3: Negative log likelihood (outside the parenthesis) and average time per epoch (in the parenthesis) for EDM and EDM+Reg for each split of DW4 and LJ13. We denote the loss 'Nan' if the training blows up and denote the unavailable result by '−'.

### 5.2.2 QM9

We further compare the four EDM models and some other most recent models, including GEOLDM [46], GRAPHLDM [46], Bridge [43], and Bridge+Force [43], for QM9. We train EDM models using Adam with learning rate $1e$-4, weight decay $1e$-12, and batch size 64. The regularization parameter is $1e$-4. We use 1000 diffusion steps and train for 1200 epochs for convergence. The four

EDM models all consist of 9 EGNN blocks and each has 6 EGCL layers and 1 time coordinates equivariant update, 256 features per layer, and using SiLU activation. Following [18], we rescale the distance between the latent variable distribution and the corresponding target normal distribution at each diffusion time during the diffusion process to make the training more stable and efficient.

**Performance and computational cost.** Figure 7 plots the training and validation loss vs. epochs; With rescaling, the training loss achieves a very small variance while the validation loss has a large variation and thus we use a moving average (over ten). We observe that the regularized models outperform the baselines. The training of EDM(unnorm) blows up quickly; in contrast, EDM(unnorm)+Reg performs the best among all models.

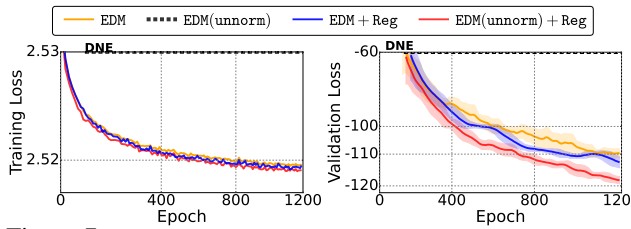

Figure 7: Training and validation loss of EDM, EDM(unnorm), EDM+Reg, and EDM(unnorm)+Reg for QM9. DNE represents the point at which the model blows up.

Table 4 compares four EDM models and four other diffusion-based models for QM9 generation. We compare the test NLL, the atomic and molecular stability estimation of 1000 molecules drawn from the different models, and the computational time. The results in Table 4 show that regularization can improve the baseline model by a significant margin in test NLL and atomic and molecular stability. EDM(unnorm)+Reg performs the best among all four EDM-related models. Though the training of regularized models requires some more computational time than the unregularized model, the computational overhead is not significant compared to the performance gain. EDM(unnorm)+Reg even outperforms all models in atomic stability and performs second in molecular stability. EDM(unnorm)+Reg does not outperform GEOLDM in molecular stability but it takes only slightly more than half of the computational cost of the latter. In Figure 15, in the appendix, we demonstrate a molecular generation process using EDM(unnorm)+Reg.

| Metrics | Test NLL ($\downarrow$) | Atomic Stab. ($\uparrow$) | Mol Stab. ($\uparrow$) | Time/Epoch(s) ($\downarrow$) |
|---|---|---|---|---|
| ENF | $-59.7$ | $85_{\pm 0.1}\%$ | $4.9_{\pm 0.2}\%$ | $4716_{\pm 540}$ |
| ENF+Reg | $\mathbf{-70.2_{\pm 2.1}}$ | $\mathbf{88_{\pm 0.1}}\%$ | $\mathbf{5.5_{\pm 0.2}}\%$ | $\mathbf{3348_{\pm 360}}$ |
| EDM(Unnorm)+Reg | $\mathbf{-123.13_{\pm 1.8}}$ | $\mathbf{98.82_{\pm 0.1}}\%$ | $85.28_{\pm 0.5}\%$ | $230_{\pm 10}$ |
| EDM(Unnorm) | Nan | Nan | Nan | Nan |
| EDM+Reg | $-114.40_{\pm 1.5}$ | $98.78_{\pm 0.1}\%$ | $83.95_{\pm 0.5}\%$ | $230_{\pm 10}$ |
| EDM | $-110.92_{\pm 1.5}$ | $98.73_{\pm 0.1}\%$ | $82.11_{\pm 0.4}\%$ | $165_{\pm 10}$ |
| GEOLDM [46] | – | $98.73\%$ | $\mathbf{89.40_{\pm 0.5}}\%$ | $425_{\pm 5}$ |
| GRAPHLDM [46] | – | $97.90\%$ | $70.50_{\pm 0.5}\%$ | $415_{\pm 5}$ |
| Bridge [43] | – | $98.7_{\pm 0.1}\%$ | $81.8_{\pm 0.2}\%$ | – |
| Bridge + Force [43] | – | $98.8_{\pm 0.1}\%$ | $84.6_{\pm 0.3}\%$ | – |
| Data | – | $99\%$ | $95.2\%$ | – |

Table 4: Contrasting the performance of different models for QM9 molecular generation. We compare the performance of different models in the test negative log-likelihood (NLL), atomic and molecular stability, and time per epoch. All experiments are running for over five random seeds. The atomic and molecular stability results of GEOLDM, GRAPHLDM, Bridge, and Bridge+Force are adapted from the original paper. We report the computational time of GEOLDM and GRAPHLDM by running the code released by the author of [46]. The results of ENF is taken from [35].
.

## 6 CONCLUDING REMARKS

In this paper, we perform a sensitivity analysis of the backpropagation of EGNN and identify a cause of the instability issue in learning ENF and EDM. Inspired by our theory, we propose a simple yet effective regularization to stabilize ENF and EDM. In addition, the regularization can improve the computational efficiency of ENF and the performance of EDM by allowing more flexible coordinate updates. We showcase the practical benefits of the proposed regularization scheme on various molecule generation benchmark tasks. There are several avenues for future work: First, developing Euclidean equivariant generative models using other equivariant GNNs. Second, developing a theoretical understanding of why the proposed regularization can stabilize forward propagation of ENF and EDM even using EGNN with the unnormalized coordinate update. Third, the training instability in ENF and EDM we study comes from the EGNN itself. Studying the efficacy of the proposed regularization for other EGNN application settings can be an interesting future work.

ETHICS STATEMENT

This paper focuses on understanding the instability issues of recently proposed equivariant generative models for molecular modeling. Based on our analysis, we propose a simple yet effective regularization to stabilize and improve the performance of the ENF and EDM model for molecular generation. We do not see any potential ethical issues in our research.

REPRODUCIBILITY STATEMENT

In pursuit of reproducible research, we have included comprehensive derivations to ease readers and we have submitted the code in the supplementary materials, along with detailed documentation, to ensure the experimental results can be easily reproduced.

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

# A  TECHNICAL PROOFS

First, we have the following lemma.

**Lemma 1.** *The following results hold:*

$$
\frac{\partial[(\boldsymbol{x}_i^l - \boldsymbol{x}_j^l)\phi_x(\boldsymbol{m}_{ij}^l)]}{\partial \boldsymbol{x}_i^l} = \phi_x(\boldsymbol{m}_{ij}^l)\boldsymbol{I}_3 + \frac{\partial \phi_x(\boldsymbol{m}_{ij}^l)}{\partial\|\boldsymbol{x}_i^l - \boldsymbol{x}_j^l\|^2} \cdot 2(\boldsymbol{x}_i^l - \boldsymbol{x}_j^l)(\boldsymbol{x}_i^l - \boldsymbol{x}_j^l)^\top \ and
$$
$$
\frac{\partial[(\boldsymbol{x}_i^l - \boldsymbol{x}_j^l)\phi_x(\boldsymbol{m}_{ij}^l)]}{\partial \boldsymbol{x}_j^l} = -\frac{\partial[(\boldsymbol{x}_i^l - \boldsymbol{x}_j^l)\phi_x(\boldsymbol{m}_{ij}^l)]}{\partial \boldsymbol{x}_i^l}.
$$
(8)

*Proof.* One can check the lemma follows from the computation below:

$$
\begin{aligned}
\frac{\partial[(\boldsymbol{x}_i^l - \boldsymbol{x}_j^l)\phi_x(\boldsymbol{m}_{ij}^l)]}{\partial \boldsymbol{x}_i^l} &= \frac{\partial(\boldsymbol{x}_i^l - \boldsymbol{x}_j^l)}{\partial \boldsymbol{x}_i^l}\phi_x(\boldsymbol{m}_{ij}^l) + (\boldsymbol{x}_i^l - \boldsymbol{x}_j^l)\frac{\partial \phi_x(\boldsymbol{m}_{ij}^l)}{\partial \boldsymbol{x}_i^l} \\
&= \boldsymbol{I}_3 \cdot \phi_x(\boldsymbol{m}_{ij}^l) + (\boldsymbol{x}_i^l - \boldsymbol{x}_j^l)\frac{\partial \phi_x(\boldsymbol{m}_{ij}^l)}{\partial\|\boldsymbol{x}_i^l - \boldsymbol{x}_j^l\|^2}\frac{\partial\|\boldsymbol{x}_i^l - \boldsymbol{x}_j^l\|^2}{\partial \boldsymbol{x}_i^l} \\
&= \phi_x(\boldsymbol{m}_{ij}^l)\boldsymbol{I}_3 + \frac{\partial \phi_x(\boldsymbol{m}_{ij}^l)}{\partial\|\boldsymbol{x}_i^l - \boldsymbol{x}_j^l\|^2}\cdot(\boldsymbol{x}_i^l - \boldsymbol{x}_j^l)\frac{\partial\|\boldsymbol{x}_i^l - \boldsymbol{x}_j^l\|^2}{\partial \boldsymbol{x}_i^l} \\
&= \phi_x(\boldsymbol{m}_{ij}^l)\boldsymbol{I}_3 + \frac{\partial \phi_x(\boldsymbol{m}_{ij}^l)}{\partial\|\boldsymbol{x}_i^l - \boldsymbol{x}_j^l\|^2}\cdot2(\boldsymbol{x}_i^l - \boldsymbol{x}_j^l)(\boldsymbol{x}_i^l - \boldsymbol{x}_j^l)^\top
\end{aligned}
$$
(9)

$$
\begin{aligned}
\frac{\partial[(\boldsymbol{x}_i^l - \boldsymbol{x}_j^l)\phi_x(\boldsymbol{m}_{ij}^l)]}{\partial \boldsymbol{x}_j^l} &= \frac{\partial(\boldsymbol{x}_i^l - \boldsymbol{x}_j^l)}{\partial \boldsymbol{x}_j^l}\phi_x(\boldsymbol{m}_{ij}^l) + (\boldsymbol{x}_i^l - \boldsymbol{x}_j^l)\frac{\partial \phi_x(\boldsymbol{m}_{ij}^l)}{\partial \boldsymbol{x}_j^l} \\
&= -\boldsymbol{I}_3 \cdot \phi_x(\boldsymbol{m}_{ij}^l) + (\boldsymbol{x}_i^l - \boldsymbol{x}_j^l)\frac{\partial \phi_x(\boldsymbol{m}_{ij}^l)}{\partial\|\boldsymbol{x}_i^l - \boldsymbol{x}_j^l\|^2}\frac{\partial\|\boldsymbol{x}_i^l - \boldsymbol{x}_j^l\|^2}{\partial \boldsymbol{x}_j^l} \\
&= -\phi_x(\boldsymbol{m}_{ij}^l)\boldsymbol{I}_3 + \frac{\partial \phi_x(\boldsymbol{m}_{ij}^l)}{\partial\|\boldsymbol{x}_i^l - \boldsymbol{x}_j^l\|^2}\cdot(\boldsymbol{x}_i^l - \boldsymbol{x}_j^l)\frac{\partial\|\boldsymbol{x}_i^l - \boldsymbol{x}_j^l\|^2}{\partial \boldsymbol{x}_j^l} \\
&= -\phi_x(\boldsymbol{m}_{ij}^l)\boldsymbol{I}_3 - \frac{\partial \phi_x(\boldsymbol{m}_{ij}^l)}{\partial\|\boldsymbol{x}_i^l - \boldsymbol{x}_j^l\|^2}\cdot2(\boldsymbol{x}_i^l - \boldsymbol{x}_j^l)(\boldsymbol{x}_i^l - \boldsymbol{x}_j^l)^\top
\end{aligned}
$$
(10)

$\square$

*Proof of Proposition 1.* Firstly,

$$
\frac{\partial \boldsymbol{h}_i^{l+1}}{\partial \boldsymbol{h}_j^l} = \frac{\partial \phi_h(\boldsymbol{h}_i^l, \boldsymbol{m}_i^l)}{\partial \boldsymbol{h}_j^l} = \frac{\partial \phi_h(\boldsymbol{h}_i^l|\boldsymbol{m}_i^l)}{\partial \boldsymbol{h}_i^l}\frac{\partial \boldsymbol{h}_i^l}{\partial \boldsymbol{h}_j^l} + \frac{\partial \phi_h(\boldsymbol{m}_i^l|\boldsymbol{h}_i^l)}{\partial \boldsymbol{m}_i^l}\frac{\partial \boldsymbol{m}_i^l}{\partial \boldsymbol{h}_j^l}
$$
(11)

If $j = i$, we have

$$
\frac{\partial \boldsymbol{h}_i^{l+1}}{\partial \boldsymbol{h}_i^l} = \frac{\partial \phi_h(\boldsymbol{h}_i^l|\boldsymbol{m}_i^l)}{\partial \boldsymbol{h}_i^l} + \frac{\partial \phi_h(\boldsymbol{m}_i^l|\boldsymbol{h}_i^l)}{\partial \boldsymbol{m}_i^l}\frac{\partial \boldsymbol{m}_i^l}{\partial \boldsymbol{h}_i^l} = \frac{\partial \phi_h(\boldsymbol{h}_i^l|\boldsymbol{m}_i^l)}{\partial \boldsymbol{h}_i^l} + \frac{\partial \phi_h(\boldsymbol{m}_i^l|\boldsymbol{h}_i^l)}{\partial \boldsymbol{m}_i^l}\sum_{k\in\mathcal{N}(i)}\frac{\partial \boldsymbol{m}_{ik}^l}{\partial \boldsymbol{h}_i^l}
$$
(12)

If $j \neq i$, then we have

$$
\frac{\partial \boldsymbol{h}_i^{l+1}}{\partial \boldsymbol{h}_j^l} = \frac{\partial \phi_h(\boldsymbol{m}_i^l|\boldsymbol{h}_i^l)}{\partial \boldsymbol{m}_i^l}\frac{\partial \boldsymbol{m}_i^l}{\partial \boldsymbol{h}_j^l} = \frac{\partial \phi_h(\boldsymbol{m}_i^l|\boldsymbol{h}_i^l)}{\partial \boldsymbol{m}_i^l}\sum_{k\in\mathcal{N}(i)}\frac{\partial \boldsymbol{m}_{ik}^l}{\partial \boldsymbol{h}_j^l} = \frac{\partial \phi_h(\boldsymbol{m}_i^l|\boldsymbol{h}_i^l)}{\partial \boldsymbol{m}_i^l}\frac{\partial \boldsymbol{m}_{ij}^l}{\partial \boldsymbol{h}_j^l}
$$
(13)

Secondly,

$$
\frac{\partial \boldsymbol{h}_i^{l+1}}{\partial \boldsymbol{x}_j^l} = \frac{\partial \phi_h(\boldsymbol{h}_i^l, \boldsymbol{m}_i^l)}{\partial \boldsymbol{x}_j^l} = \frac{\partial \phi_h(\boldsymbol{m}_i^l|\boldsymbol{h}_i^l)}{\partial \boldsymbol{m}_i^l}\frac{\partial \boldsymbol{m}_i^l}{\partial \boldsymbol{x}_j^l} = \frac{\partial \phi_h(\boldsymbol{m}_i^l|\boldsymbol{h}_i^l)}{\partial \boldsymbol{m}_i^l}\sum_{k\in\mathcal{N}(i)}\frac{\partial \boldsymbol{m}_{ik}^l}{\partial \boldsymbol{x}_j^l}
$$
(14)

If $j = i$, we have

$$
\begin{aligned}
\frac{\partial \boldsymbol{h}_i^{l+1}}{\partial \boldsymbol{x}_i^l} &= \frac{\partial \phi_h(\boldsymbol{m}_i^l | \boldsymbol{h}_i^l)}{\partial \boldsymbol{m}_i^l} \sum_{k \in \mathcal{N}(i)} \frac{\partial \boldsymbol{m}_{ik}^l}{\partial \boldsymbol{x}_i^l} \\
&= \frac{\partial \phi_h(\boldsymbol{m}_i^l | \boldsymbol{h}_i^l)}{\partial \boldsymbol{m}_i^l} \sum_{k \in \mathcal{N}(i)} \frac{\partial \boldsymbol{m}_{ik}^l}{\partial \|\boldsymbol{x}_i^l - \boldsymbol{x}_k^l\|^2} \frac{\partial \|\boldsymbol{x}_i^l - \boldsymbol{x}_k^l\|^2}{\partial \boldsymbol{x}_i^l} \\
&= \frac{\partial \phi_h(\boldsymbol{m}_i^l | \boldsymbol{h}_i^l)}{\partial \boldsymbol{m}_i^l} \sum_{k \in \mathcal{N}(i)} \frac{\partial \boldsymbol{m}_{ik}^l}{\partial \|\boldsymbol{x}_i^l - \boldsymbol{x}_k^l\|^2} \cdot 2(\boldsymbol{x}_i^l - \boldsymbol{x}_k^l)^\top
\end{aligned}
\tag{15}
$$

If $j \neq i$, then we have

$$
\begin{aligned}
\frac{\partial \boldsymbol{h}_i^{l+1}}{\partial \boldsymbol{x}_j^l} &= \frac{\partial \phi_h(\boldsymbol{m}_i^l | \boldsymbol{h}_i^l)}{\partial \boldsymbol{m}_i^l} \sum_{k \in \mathcal{N}(i)} \frac{\partial \boldsymbol{m}_{ik}^l}{\partial \boldsymbol{x}_j^l} \\
&= \frac{\partial \phi_h(\boldsymbol{m}_i^l | \boldsymbol{h}_i^l)}{\partial \boldsymbol{m}_i^l} \frac{\partial \boldsymbol{m}_{ij}^l}{\partial \boldsymbol{x}_j^l} \\
&= \frac{\partial \phi_h(\boldsymbol{m}_i^l | \boldsymbol{h}_i^l)}{\partial \boldsymbol{m}_i^l} \frac{\partial \boldsymbol{m}_{ij}^l}{\partial \|\boldsymbol{x}_i^l - \boldsymbol{x}_j^l\|^2} \frac{\partial \|\boldsymbol{x}_i^l - \boldsymbol{x}_j^l\|^2}{\partial \boldsymbol{x}_j^l} \\
&= -\frac{\partial \phi_h(\boldsymbol{m}_i^l | \boldsymbol{h}_i^l)}{\partial \boldsymbol{m}_i^l} \frac{\partial \boldsymbol{m}_{ik}^l}{\partial \|\boldsymbol{x}_i^l - \boldsymbol{x}_j^l\|^2} \cdot 2(\boldsymbol{x}_i^l - \boldsymbol{x}_j^l)^\top
\end{aligned}
\tag{16}
$$

Thirdly,

$$
\frac{\partial \boldsymbol{x}_i^{l+1}}{\partial \boldsymbol{h}_j^l} = \sum_{k \in \mathcal{N}(i)} \frac{\partial [(\boldsymbol{x}_i^l - \boldsymbol{x}_k^l) \phi_x(\boldsymbol{m}_{ik}^l)]}{\partial \boldsymbol{h}_j^l} = \sum_{k \in \mathcal{N}(i)} (\boldsymbol{x}_i^l - \boldsymbol{x}_k^l) \frac{\partial \phi_x(\boldsymbol{m}_{ik}^l)}{\partial \boldsymbol{h}_j^l}
\tag{17}
$$

If $j = i$, then we have

$$
\frac{\partial \boldsymbol{x}_i^{l+1}}{\partial \boldsymbol{h}_i^l} = \sum_{k \in \mathcal{N}(i)} (\boldsymbol{x}_i^l - \boldsymbol{x}_k^l) \frac{\partial \phi_x(\boldsymbol{m}_{ik}^l)}{\partial \boldsymbol{h}_i^l} = \sum_{k \in \mathcal{N}(i)} (\boldsymbol{x}_i^l - \boldsymbol{x}_k^l) \frac{\partial \phi_x(\boldsymbol{m}_{ik}^l)}{\partial \boldsymbol{m}_{ik}^l} \frac{\partial \boldsymbol{m}_{ik}^l}{\partial \boldsymbol{h}_i^l}
\tag{18}
$$

If $j \neq i$, then we have

$$
\frac{\partial \boldsymbol{x}_i^{l+1}}{\partial \boldsymbol{h}_j^l} = (\boldsymbol{x}_i^l - \boldsymbol{x}_j^l) \frac{\partial \phi_x(\boldsymbol{m}_{ij}^l)}{\partial \boldsymbol{h}_j^l} = (\boldsymbol{x}_i^l - \boldsymbol{x}_j^l) \frac{\partial \phi_x(\boldsymbol{m}_{ij}^l)}{\partial \boldsymbol{m}_{ij}^l} \frac{\partial \boldsymbol{m}_{ij}^l}{\partial \boldsymbol{h}_j^l}
\tag{19}
$$

Finally, notice that

$$
\frac{\partial \boldsymbol{x}_i^{l+1}}{\partial \boldsymbol{x}_j^l} = \frac{\partial \boldsymbol{x}_i^l}{\partial \boldsymbol{x}_j^l} + \sum_{k \in \mathcal{N}(i)} \frac{\partial [(\boldsymbol{x}_i^l - \boldsymbol{x}_k^l) \phi_x(\boldsymbol{m}_{ik}^l)]}{\partial \boldsymbol{x}_j^l}
\tag{20}
$$

It is evident that $\frac{\partial \boldsymbol{x}_i^l}{\partial \boldsymbol{x}_j^l} = \boldsymbol{I}_3$ if $j = i$ and $0$ if $j \neq i$ and $\frac{\partial [(\boldsymbol{x}_i^l - \boldsymbol{x}_k^l) \phi_x(\boldsymbol{m}_{ik}^l)]}{\partial \boldsymbol{x}_j^l} = 0$ if $j \neq i$ or $k$. Moreover, applying Lemma 1 to (20), we obtain the following result:

$$
\frac{\partial \boldsymbol{x}_i^{l+1}}{\partial \boldsymbol{x}_j^l} = \begin{cases} \boldsymbol{I}_3 + \sum_{k \in \mathcal{N}(i)} \boldsymbol{G}_{ik}^l & \text{if } j = i, \\ -\boldsymbol{G}_{ij}^l & \text{if } j \neq i. \end{cases}
\tag{21}
$$

$\square$

## A.1 THE GRADIENT OF THE LOSS WITH RESPECT TO RELATIVE DISTANCES

In this subsection, we verify that the gradient of the loss with respect to the pairwise distance explodes when the training blows up. Figure 8 plots epochs vs. $\partial \mathcal{L} / \partial \|\boldsymbol{x}^i - \boldsymbol{x}^j\|$ for the DW4 task with 100 training data. We observe that when vanilla EGNN is used, the norm of the gradient $\partial \mathcal{L} / \partial \|\boldsymbol{x}^i - \boldsymbol{x}^j\|$ explodes rapidly. Using normalized coordinate updates can mitigate the gradient explosion and regularization can further stabilize the training process.

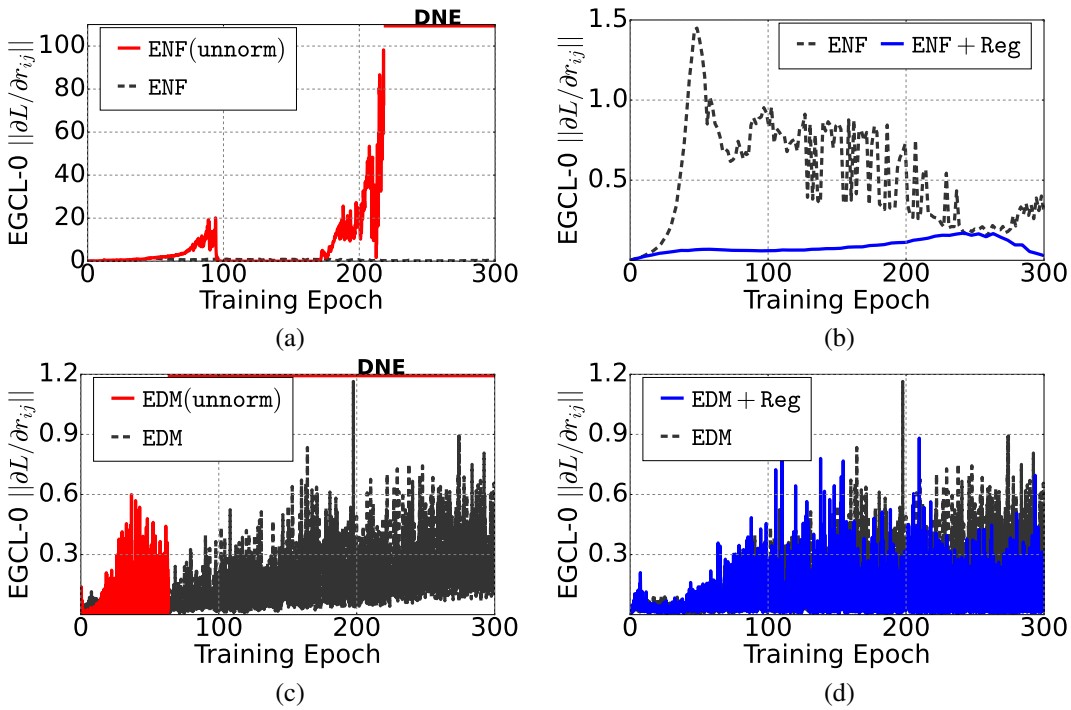

Figure 8: The gradient of the loss $\mathcal{L}$ with respect to the relative distances $r_{ij} = \|\boldsymbol{x}_i - \boldsymbol{x}_j\|$ for the 0-th EGCL layer for ENF in (a) and (b) and for EDM in (c) and (d) when training on the DW4 task with 100 training molecules. We observe that training with regularization provides the most stability in the loss of training.

## B  ADDITIONAL EXPERIMENTAL RESULTS AND EXPERIMENTAL DETAILS

### B.1  ADDITIONAL DW4 RESULTS.

In Figure 9, we compare the sampled and ground truth energy probability density functions for both ENF and ENF+Reg using the DW4 dataset. The results show that the sampled distribution by using ENF+Reg is closer to the ground truth than the sampled distribution using ENF. Figure 10 further compares the mean relative distance of the samples generated by ENF and ENF+Reg with the ground truth, showing that ENF+Reg can generate samples that are more in proximity to the ground truth.

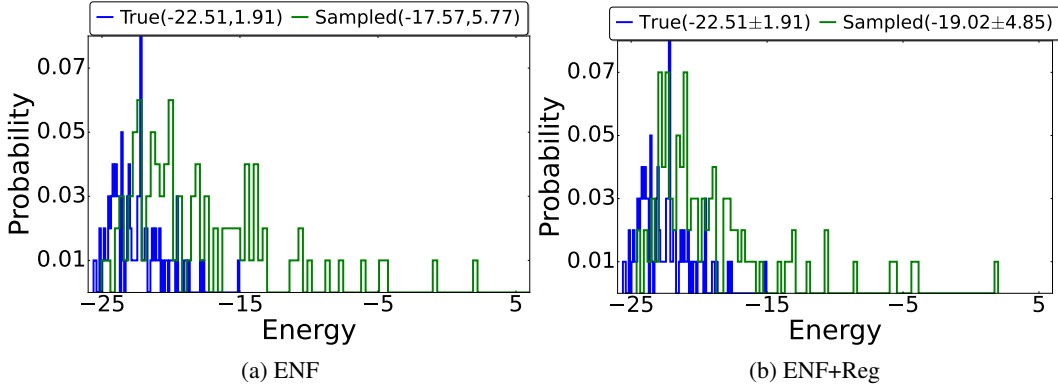

Figure 9: Probability distributions (mean±standard deviation) of the molecular energy for 100 true and sampled molecules on the DW4 dataset using ENF and ENF+Reg with 100 training samples.

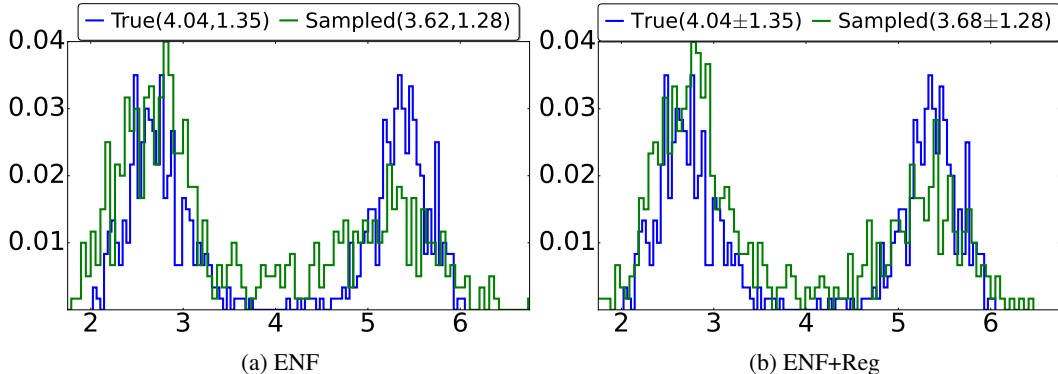

(a) ENF        (b) ENF+Reg

Figure 10: Probability distributions (mean±standard deviation) of the relative distances for 100 true and sampled molecules on the DW4 dataset using ENF and ENF+Reg with 100 training samples.

## B.2 Additional LJ13 results

In Figure 11, we compare the sampled and ground truth energy probability density functions for both ENF and ENF+Reg using the LJ13 dataset. The results show that the sampled distribution by using ENF+Reg is closer to the ground truth than the sampled distribution using ENF.

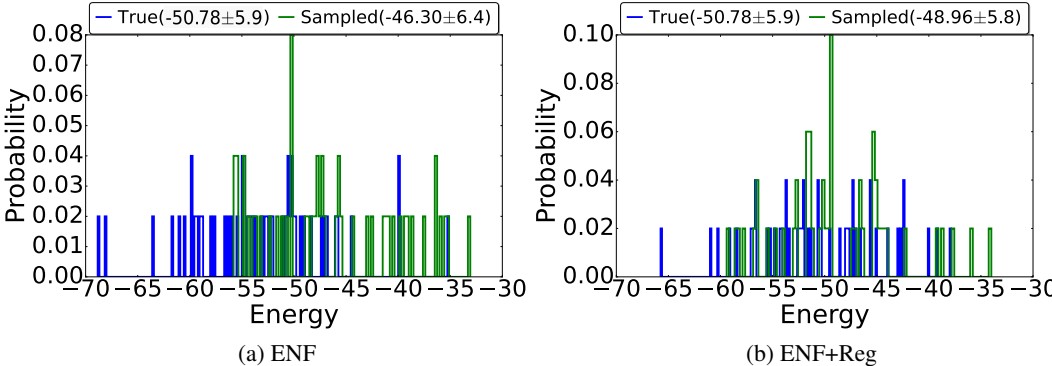

(a) ENF        (b) ENF+Reg

Figure 11: Probability distributions (mean±standard deviation) of the molecular energy for 100 true and sampled molecules on the LJ13 dataset using ENF and ENF+Reg with 100 training samples.

## B.3 Regularization vs. gradient clipping and gradient penalty

In this subsection, we compare the performance of the proposed regularization against other commonly used techniques to stabilize the training, including gradient clipping and gradient penalty. We compare the performance of three approaches in the context of both ENF and EDM in the DW4 generation task.

### B.3.1 Gradient clipping

Gradient clipping [47] involves capping the error derivatives before propagating them back through the network. The capped gradients are used to update the weights, resulting in smaller weight updates and stabilizing the training. In our experiments, we clip the gradient using the following standard formula:

$$grad = \frac{\max\{\alpha, ||grad||\}}{||grad||} * grad \tag{22}$$

with clipping threshold

$$\alpha = QueueNorm_{mean} + \beta * QueueNorm_{std} \tag{23}$$

where $grad$ is the gradient to be clipped, $\beta \geq 0$ is the clipping parameter, and $QueueNorm$ denotes the norm of the gradients in the clipping queue in the training. Figure 12 and Table 5 compare

the performance of training ENF on the DW4 dataset—with the training set of size 100—using gradient clipping with various hyperparameters $\beta$ and our proposed regularization. These results show that the proposed regularization outperforms gradient clipping by a remarkable margin in terms of stabilizing the training, reducing the computational time, and improving the generation quality.

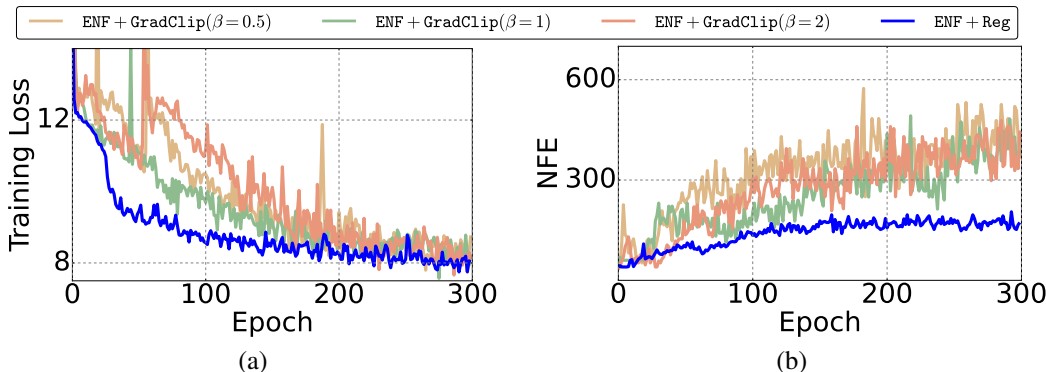

Figure 12: Comparing training ENF on the DW4 datasets (with 100 training data) using our proposed regularization against gradient clipping (with various hyperparameters $\beta$). (a) shows that regularization enables more stable training of ENF than using gradient clipping, and (b) shows that regularization enables faster training of ENF than using gradient clipping.

| Split | Test NLL($\downarrow$) | Time/Epoch($\downarrow$, second) |
|---|---|---|
| ENF+Reg | $\mathbf{12.02}_{\pm 0.2}$ | $\mathbf{8.6}$ |
| ENF+GradClip($\beta = 0.5$) | $12.93_{\pm 0.1}$ | 21.3 |
| ENF+GradClip($\beta = 1$) | $12.67_{\pm 0.1}$ | 19.5 |
| ENF+GradClip($\beta = 2$) | $12.23_{\pm 0.3}$ | 18.7 |

Table 5: Comparing training ENF on the DW4 datasets (with 100 training data) using our proposed regularization against gradient clipping (with various hyperparameters $\beta$) in test negative log-likelihood and average time per epoch.

### B.3.2 GRADIENT PENALTY

In this subsection, we further compare the performance of our proposed regularization against the gradient penalty. In the gradient penalty, we add a scaled norm of the gradient to the loss function to force the gradient to be relatively small to stabilize the training. The loss function with gradient penalty can be written as follows:

$$\tilde{\mathcal{L}} = \mathcal{L} + \gamma \left\| \frac{\partial \mathcal{L}}{\partial \theta} \right\|, \tag{24}$$

where $\mathcal{L}$ is the loss function, $\theta$ is the weights of the model under training, and $\gamma$ is the penalty parameter.

**ENF on DW4**: Figure 13 and Table 6 compare the performance of training ENF, using our proposed regularization against gradient penalty with various $\gamma$, on the DW4 dataset with training size 100.

| Split | Test NLL($\downarrow$) | Time/Epoch($\downarrow$, second) |
|---|---|---|
| ENF+Reg | $\mathbf{12.02}_{\pm 0.2}$ | $\mathbf{8.6}$ |
| ENF+GradPenalty($\gamma = 1e-3$) | $12.24_{\pm 0.2}$ | 22.7 |
| ENF+GradPenalty($\gamma = 1e-4$) | $12.53_{\pm 0.2}$ | 28.6 |
| ENF+GradPenalty($\gamma = 1e-5$) | $12.39_{\pm 0.3}$ | 17.9 |

Table 6: Comparing ENF+Reg against ENF+Gradient penalty in test negative log-likelihood and average time per epoch in training process on the DW4 with 100 training samples.

**EDM on DW4**: Figure 14 and Table 7 compares the performance of EDM—training using our proposed regularization or gradient penalty—on the DW4 dataset with training size 1000.

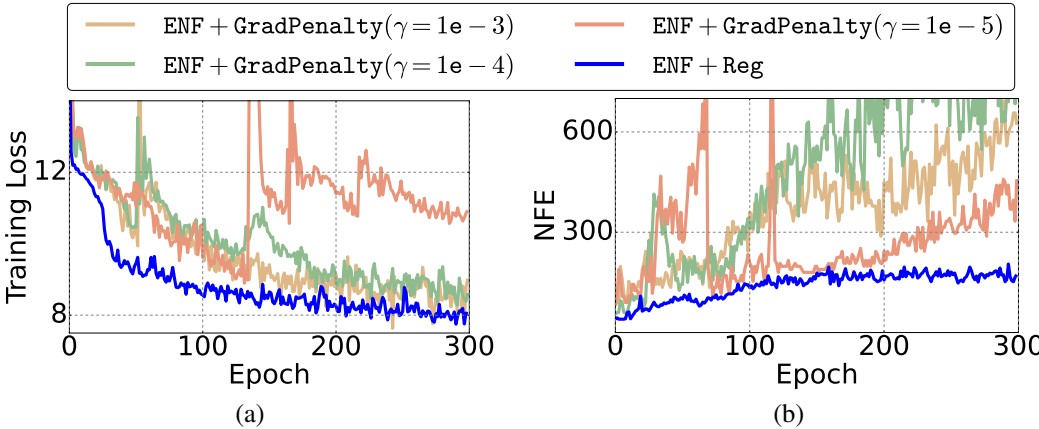

Figure 13: Comparing ENF+Reg against ENF+Gradient penalty in (a) training loss and (b) the number of function evaluations (NFEs) per training epoch used by the ODE solver on DW4 with 100 training data.

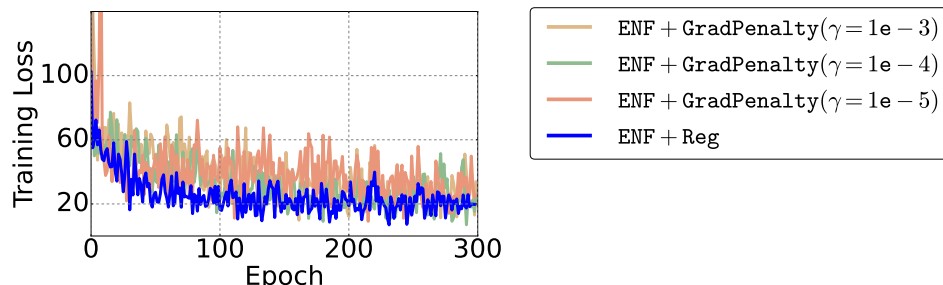

Figure 14: Comparing EDM+Reg against EDM+Gradient penalty in training loss on DW4 with 1000 training data.

| Split | Test NLL($\downarrow$) | Time/Epoch($\downarrow$, second) |
|---|---|---|
| ENF+Reg | $\mathbf{22.39}_{\pm 0.8}$ | $\mathbf{1.6}$ |
| ENF+GradPenalty($\gamma = 1e - 3$) | $23.15_{\pm 0.9}$ | 1.7 |
| ENF+GradPenalty($\gamma = 1e - 4$) | $22.99_{\pm 0.9}$ | 1.7 |
| ENF+GradPenalty($\gamma = 1e - 5$) | $24.52_{\pm 1.2}$ | 1.7 |

Table 7: Comparing EDM+Reg against EDM+Gradient penalty in test negative log-likelihood on DW4 with 1000 training data.

### B.4 Additional dataset details

**DW4.** The DW4 dataset contains molecules with 4 atoms each in 2-dimensional space. Node information is a one-hot embedding of the atom, while edges denote atomic bonds. We train on two sets of data, one with 100 training molecules and a second with 1000 training molecules.

**LJ13.** The LJ13 dataset contains molecules with 13 atoms each in 3-dimensional space. Node information is a one-hot embedding of the atom, while edges denote atomic bonds. We train on two sets of data, one with 100 training molecules and the other with 1000 training molecules.

**QM9.** The QM9 dataset consists of a collection of molecules containing up to nine heavy atoms, including carbon (C), oxygen (O), nitrogen (N), and sulfur (S). Atom features include a one-hot encoding of the nodes as well as several additional atomic properties. Edges encode bond information. This dataset is commonly used for molecular property prediction and graph classification. We use two subsets of the data for training the molecular generation task.

## B.5 A SAMPLE MOLECULE GENERATED BY EDM

Figure 15 shows the selected molecules generated by our regularized EDM and also shows the process of how it generates a molecule from standard normal noise, which indicates that our model is able to generate stable molecules.

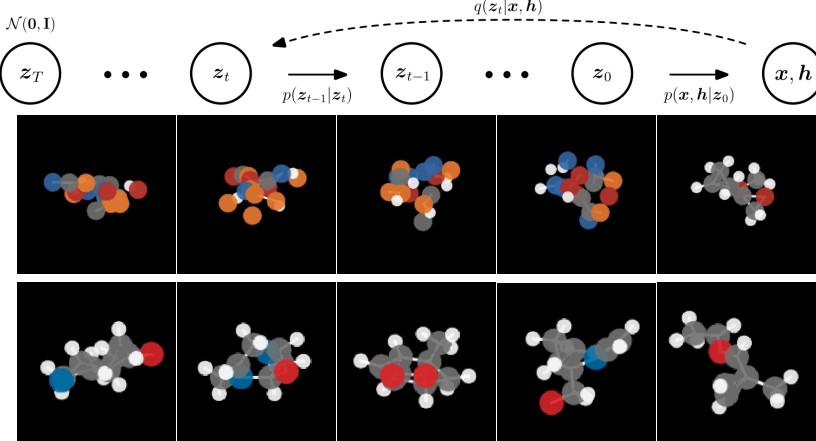

Figure 15: Overview of EDM(unnorm)+Reg on QM9. The upper figures show the generating process. To generate molecules, coordinates $x$ and features $h$ are generated by denoising variables $z_t$ starting from standard normal noise $z_T$. The lower figures show the selected samples generated by the denoising process of our EDM(unnorm)+Reg trained on QM9.

