# OpenReview forum: "Stabilized E(n)-Equivariant Graph Neural Networks-assisted Generative Models"
_ICLR.cc/2024/Conference — Submitted to ICLR 2024_

### Official Review · Reviewer_Tc5b · 2023-10-30

**Soundness:** 3 good
**Presentation:** 3 good
**Contribution:** 2 fair
**Rating:** 5
**Confidence:** 4

**Summary:**

This work proposes a regularizer to alleviate the training instability issue of EGNN in generative modeling. It analyzes the gradient of coordinate updates with respect to input and figures out the sensitive term that results in numerical explosions. Experimental results show that the proposed regularizer can successfully stabilize the ENF and EDM training.

**Strengths:**

(1) This is the first time delving into the numerical instability issue of the ENF and EDM training. This is a very important challenge and fundamental research question that was overlooked in the previous research. This reviewer agrees with the significance of the proposed challenge;

(2) The writing is basically well-organized, containing both theoretical analysis and experimental analysis, which makes the paper technically solid;

**Weaknesses:**

(1) This work analyzes the gradient update of a single-layer EGNN but this reviewer does not see an analysis of special reasons leading to unstable ENF and EDM training. Only a single sentence states that “In both ENF and EDM, the graph node coordinates keep changing during the generative process, and abnormal coordinate updates may occur.” However, the generative process is the inference process while this paper is dealing with the training instability. Therefore, the ENF and EDM training instability is not accurately summarized. Is EGNN on the given training data stable? Is ENF and EDM unstable on the given training data? If the answers to both questions are yes, then this might indicate that there are special reasons resulting in the instability of ENF and EDM training. If ENF and EDM training instability is caused by EGNN training only (no special stuff), then the discussions over the ENF and EDM are redundant and not necessary. In general, this reviewer thinks the preliminary discussions over the training instability are not sufficient.

(2) The proposed regularizer is a sum of gradient calculations, which is extremely hard to compute. The summations are computed over all atom pairs (i, j) and EGNN layers L with at least O(N**2*L) time complexity. Hence, although the proposed regularizer is a very effective approach to restricting the gradient norm, the calculation of the proposed regularizer could not be scalable to either larger particle systems or deep EGNN architectures.

**Questions:**

(1) What is the time complexity of the calculation of the proposed regularizer? How could the proposed approach scale to larger particle systems and deep EGNN architectures?

(2) What are the special reasons for numerical instabilities in ENF and EDM training? Or the instability of ENF and EDM is all caused by EGNN?

---

> ### Author Response · Authors · 2023-11-15
> **Response to Reviewer Tc5b**
>
> Thank you for your thoughtful review and valuable feedback. Since the questions the reviewer raised align with the reviewer’s comments on the weaknesses of our paper, we provide point-by-point responses to the reviewer’s questions.
>
> ---
>
>
> **Q1. What is the time complexity of the calculation of the proposed regularizer? How could the proposed approach scale to larger particle systems and deep EGNN architectures?**
>
> **Reply:** The graphs considered in the molecular generation tasks are sparse, and therefore the time complexity of the regularization term is $O(NL)$ with $N$ and $L$ being the number of particles and the depth of EGNN, respectively.
>
> All summations are taken over the neighborhood of a given graph node, we have fixed all inappropriate notations in the revision.
>
> In our paper, we have also compared the computational time of the EDM with and without regularization, showing the extra computational time is not very significant. The computational overhead due to the regularization term is much smaller than the baseline computational time.
>
>
> ---
>
> **Q2. What are the special reasons for numerical instabilities in ENF and EDM training? Or the instability of ENF and EDM is all caused by EGNN?**
>
> **Reply:** The instability issue of ENF and EDM we study originates from the EGNN model in the training phase, and we have pointed this out in the first paragraph of Section 1.2 of our paper. We focus on ENF and EDM for the following two main reasons:
>
> - Though EGNN is a general E(n)-equivariant graph neural network, one of the most remarkable applications of EGNN is for equivariant generative modeling. If we look at the citations of the EGNN paper, a very large portion are related to equivariant generative modeling.
>
> - Besides stabilizing training ENF and EDM, the proposed regularization scheme brings additional important benefits to ENF and EDM.
> For ENF, because solving a high-dimensional ODE is required in both training and testing, the proposed regularization can also regularize the Lipschitz constant of the ODE to improve the computational efficiency of ENF. For EDM, the proposed regularization allows EGNN to use an unnormalized coordinate update, and such an unnormalized coordinate update can improve the performance of EDM.
>
> In the revision, we have further improved the discussion of the instability issues of training ENF and EDM in Section 1.2 and Section 3.
>
> Finally, in the concluding remarks section, we have added the discussion that the training instability in ENF and EDM we study comes from the EGNN itself. Studying the efficacy of the proposed regularization for other EGNN applications can be an interesting future work.
>
>
> ---
>
> We have updated our submission based on the reviewer's feedback, with the revision highlighted in blue. We are happy to address further questions on our paper. Thank you for considering our rebuttal.

---

> > ### Comment · Reviewer_Tc5b · 2023-11-23
> > **Reply to the author reponse**
> >
> > Dear Author,
> >
> > After careful reading over the response, we are still concerned about the general consistency of this paper. The most important reason that leads to the numerical instabilities of EGNN when equipped with diffusion and flow is not fully revealed in this work. Currently, only some observations prove the existence of the numerical instability issue. The gradient analysis is actually not very important in this case as its theoretical result follows the intuition. The key part is exploring the special reasons that induce the numerical explosion of EDM and ENF. What particular operations make the EDM and ENF training unstable? It seems this reason is still not clearly stated in the author's rebuttal. Therefore, we think this paper is still inconsistent and we believe this paper can be accepted in the future after rounds of revisions.
> >
> > Best.

---

> ### Author Response · Authors · 2023-11-23
> **Response**
>
> Dear Reviewer,
>
> We thank the reviewer for the further feedback. The coordinate update operations cause instability in backpropagation, resulting in highly oscillatory or even exploding gradients and making the training unstable. After observing instability empirically in training, it is natural to investigate the gradients - this is the motivation for our theoretical study.
>
> Thank you for considering our rebuttal.
>
>
> Regards,
>
> Authors

---

### Official Review · Reviewer_9sae · 2023-10-31

**Soundness:** 2 fair
**Presentation:** 3 good
**Contribution:** 2 fair
**Rating:** 3
**Confidence:** 3

**Summary:**

This paper builds upon E(n)-equivariant graph convolutional networks (EGNN) by normalizing the convolution layer with respect to the node’s positions, and by adding a regularization term to the loss that promotes small gradients during training. The goal is to stabilize the training of graph generative models: normalizing flows and diffusion models.

**Strengths:**

The paper is mostly well written and clear. The regularization and normalization methods are shown in experiments to improve performance.

**Weaknesses:**

The contribution is incremental. The contribution is just dividing by a normalization factor the well known EGCL, and adding a regularization factor to training. These are natural modifications of EGCL, and any other network, and I suspect that many practitioners apply such tricks often. The analysis of stability of backpropagation is partial, and missing important contributing terms to the magnitude of the gradients.

**Questions:**

Page 4: “In contrast, normalizing coordinates can avoid abnormal coordinate updates from large differences among coordinates of neighboring nodes” - but on the flip side, without the $+1$ regularization in the denominator, it is unstable to small coordinates. But with the $+1$ normalization, close-by nodes contribute a very small difference. How do you then choose the scale of the coordinates for the $+1$ to work well? Why do you use $+1$ and not $+b$ for some $b$ that depends on the characteristic target distance between modes?

Please explain how you construct a graph from the node locations and features.

Proposition 1: In all sums, shouldn’t you sum over the neighborhood, and not the whole graph? $m_{i,j}$ is only defined when $(i,j)$ is an edge.

Section 3.2 Sensitivity Analysis: The normalized EGCL is different from the unnormalized one. Why don’t you compute the derivatives of the normalized version if this is the method you propose? Also, it is strange to directly write the derivative of $\phi_x$ with respect to $\|x_i-x_j\|$. You need to use the chain rule, and first differentiate with respect to $m_{i,j}$.

Notations: you did not define $I_3$.

Section 4: I don’t think that the derivative $\partial f^{l+1}/\partial f^l$ is the only main contributor to the size of $\partial L (f^L)/\partial \theta$. To see this, note for example that when partitioning the parameters to the last layer parameters $\theta^L$ and the previous layers parameters $\theta^{L-1}$,  $\partial L (f^L)/\partial \theta$ has two components. First, $\partial L/\partial f^L \cdot  \partial f^L\partial \theta^L$, and then $\partial L/\partial f^L \cdot  \partial f^L\partial f^{L-1}\cdot \partial f^{L-1}\partial \theta^{L-1}$. Hence, $ \partial f^{l}/\partial \theta^{l}$ are also important, and these are roughly going to depend on $\|x_i-x_j\|^l$ by induction, as each later multiples by a factor of order $\|x_i-x_j\|$. I think that this is another main reason you would like to normalize $x_i-x_j$.

This example is to illustrate that the analysis presented in this paper is partial. There is no systematic analysis of all components that contribute to the gradient.

One thing that is confusing in the analysis is that it analyzes only the unnormalized layer.  If you want to show that the normalized layer is more stable, you should write down a gradient analysis of it also. And write the full gradient with respect to the model parameters, as these are the gradients in training.


Experiments: why are EDMs and ENF not compared against each other and other models in molecule generation in one table? I only see a comparison between EDM and other methods.

---

> ### Author Response · Authors · 2023-11-15
> **Response to Reviewer 9sae (1/2)**
>
> Thank you for your thoughtful review and valuable feedback. We fear that the reviewer ***misunderstood our proposed regularization scheme as simply adding factor 1 to the denominator of the coordinate update formula in EGCL***, i.e. Equation 1, which makes the reviewer feel our contribution is incremental. We would like to stress that our proposed regularization is given by Equation 4 rather than adding factor 1 to the denominator of the coordinate update formula in EGCL. Adding factor 1 to the denominator of the coordinate update formula in EGCL is a default in ENF and EDM, which is not our contribution. In the revision, we have significantly revised the paper to make our contribution more clear. In what follows, we provide point-by-point responses to your comments.
>
> ---
>
> **Q1. The contribution is incremental. The contribution is just dividing by a normalization factor the well known EGCL, and adding a regularization factor to training.**
>
> **Reply:** We respectfully disagree that our contribution is incremental. We fear there are some misunderstandings of our contribution. Please allow us to clarify our contribution below:
>
> - First, dividing by a normalization factor, or using the normalized coordinate update is not our contribution, which is a default use in ENF and EDM. Moreover, ***the regularization we proposed is not by adding a factor to the denominator in the normalized coordinate update***. Instead, our proposed regularization scheme is given in Equation 4 in our paper.
>
> - Second, the ***analysis of the instability issues in training ENF and EDM*** Section 3 is our first contribution. In particular, we show that the training instability remains even when using the normalized coordinate update. By performing a sensitivity analysis in Section 3.2, we identify a source of the training instability issue. In particular, the term $\frac{\partial {\bf x}\_i^{l+1}}{\partial {\bf x}\_j^l}$ is the most sensitive term to the coordinate update in EGNN during the training process.
>
> - Third, in Section 4, based on the expression of $\frac{\partial {\bf x}\_i^{l+1}}{\partial {\bf x}\_j^l}$, we propose a new regularization term to improve the training of ENF and EDM. ***The new regularization is given by $\sqrt{\sum_{i,j\in\mathcal{N}(i),l}( \frac{\partial \phi_x({\bf m}_{ij}^l) }{\partial  ||{\bf x}_i^l-{\bf x}_j^l ||^2  }  )^2 }$, which is novel and highly nontrivial.***
> Besides stabilizing training ENF and EDM, we further show that ***the proposed regularization can accelerate the training and testing of ENF and improve the performance of EDM***.
> In Section 5, we numerically verify the efficacy of the proposed regularization and confirm our theoretical results.
>
> We believe the above contributions are significant.
>
> ---
>
> **Q2. The analysis of stability of backpropagation is partial, and missing important contributing terms to the magnitude of the gradients.**
>
> **Reply:** We will address this comment later as a more concrete question on this comment has been raised by the reviewer.
>
>
> ---
>
> **Q3. Page 4: “In contrast, normalizing coordinates can avoid abnormal coordinate updates from large differences among coordinates of neighboring nodes” - but on the flip side, without the +1 regularization in the denominator, it is unstable to small coordinates. But with the +1 normalization, close-by nodes contribute a very small difference. How do you then choose the scale of the coordinates for the +1 to work well? Why do you use +1 and not +b for some b that depends on the characteristic target distance between modes?**
>
> **Reply:** +1 in the denominator is not the regularization scheme we proposed, it was used in the original ENF and EDM models. Again, our proposed regularization is given by Equation 4.
>
> ---
>
> **Q4. Please explain how you construct a graph from the node locations and features.**
>
>
> **Reply:** We follow the original ENF paper to construct the graphs for DW4 and LJ13. For QM9, the graph is given in the benchmark datasets; see reference [1] listed below for details. Specifically, the graph is padded with nodes containing zero-valued features until all molecules share the same number of nodes. Then the graph edges are constructed by connecting all nodes with non-zero features. We follow this same graph construction procedure from the baseline ENF and EDM models. The exact code snippet is provided in reference [2] listed below and is executed when the graphs are collated into mini-batches for training.
>
> [1] Anderson et al. Cormorant: Covariant Molecular Neural Networks. NeurIPS 2019.
>
> [2] https://github.com/vgsatorras/en_flows/blob/main/qm9/data/collate.py#L78C4-L83C64
>
> ---

---

> ### Author Response · Authors · 2023-11-15
> **Response to Reviewer 9sae (2/2)**
>
> **Q5. Proposition 1: In all sums, shouldn’t you sum over the neighborhood, and not the whole graph? $m_{ij}$ is only defined when $(i,j)$ is an edge.**
>
>
> **Reply:** Indeed, the sum is over the neighborhood. This has been fixed in the revision. Thank you for pointing this out.
>
> ---
>
> **Q6. Section 3.2 Sensitivity Analysis: The normalized EGCL is different from the unnormalized one. Why don’t you compute the derivatives of the normalized version if this is the method you propose? Also, it is strange to directly write the derivative of $\phi_x$ with respect to $|x_i-x_j|$. You need to use the chain rule, and first differentiate with respect to $m_{ij}$.**
>
>
> **Reply:** We first stress again that the normalized version of EGCL is not what we proposed.
>
> The notation $\frac{\partial \phi_x }{\partial ||{\bf x}_i-{\bf x}_j||^2}$ is clear while its detailed expression is obtained by using the chain rule. In what follows, we would like to provide basic examples to illustrate this.
>
> Let $w=f(z)$, $z=g(y)$, and $y=h(x)$. Though the exact expression $\frac{\partial w}{\partial x}$ is $\frac{\partial w}{ \partial z}\cdot \frac{\partial z}{ \partial y}\cdot \frac{\partial y}{ \partial x}$, in practice, people will simply write $\frac{\partial z}{\partial x}$ rather than always write down the exact expression, which is rather redundant.
>
>
>
> ---
>
> **Q7. Notations: you did not define I_3.**
>
>
> **Reply:** Thank you for pointing this out, $I_3$ is the $3\times 3$ identity matrix. We have clarified this in the revision.
>
> ---
>
> **Q8. Section 4: I don’t think that the derivative $\partial f^{l+1}/\partial f^l$ is the only main contributor to the size of $\partial L(f^L)/\partial \theta$. To see this, note for example that when partitioning the parameters to the last layer parameters $\theta^L$ and the previous layers parameters $\theta^{L-1}$, $\partial L(f^L)/\partial \theta$ has two components. First, $\partial L/\partial f^L\cdot \partial f^L\partial \theta^L$, and then $\partial L/\partial f^L\cdot \partial f^L\partial f^{L-1}\cdot \partial f^{L-1}\partial\theta^{L-1}$. Hence, $\partial f^l/\partial \theta^l$ are also important, and these are roughly going to depend on $|x_i-x_j|^l$ by induction, as each later multiples by a factor of order $|x_i-x_j|$. I think that this is another main reason you would like to normalize $x_i-x_j$. This example is to illustrate that the analysis presented in this paper is partial. There is no systematic analysis of all components that contribute to the gradient.**
>
>
> **Reply:** We respectfully disagree that our analysis is partial. Notice that in Equation 6 of our paper, we have clearly stated that ***$\theta$ is the weight parameter appearing at the $l$-th layer of EGNN.***
>
> We would like to point out again that our sensitivity analysis aims to identify the most sensitive term with respect to $||{\bf x}_i-{\bf x}_j||$ in the gradient $\frac{\partial L}{\partial \theta}$. Through this sensitivity analysis, we propose the regularization scheme in Equation 4.
>
> ---
>
> **Q9. One thing that is confusing in the analysis is that it analyzes only the unnormalized layer. If you want to show that the normalized layer is more stable, you should write down a gradient analysis of it also. And write the full gradient with respect to the model parameters, as these are the gradients in training.**
>
>
> **Reply:** We believe there is a misunderstanding, we are not aiming to show the normalized layer is more stable than the unnormalized layer. ***What we are showing is that using no matter normalized or unnormalized coordinate updates, training ENF and EDM suffers from the instabilities issue.***
>
> In our paper, due to the page limit, we only write down the results for the unnormalized coordinate update. For the normalized coordinate update, a similar result can be derived by merging the normalization factor into $\phi_x({\bf m}_{ij}^l)$ in Equation 5.
>
>
> ---
>
> **Q10. Experiments: why are EDMs and ENF not compared against each other and other models in molecule generation in one table? I only see a comparison between EDM and other methods.**
>
> **Reply:** In our original submission, we separated the results of ENF and EDM into two tables since ENF (normalizing flows) and EDM (diffusion models) are quite different models. In the revision, we have merged the two tables.
>
> ---
>
> We have updated our submission based on the reviewer's feedback, with the revision highlighted in blue. We are happy to address further questions on our paper. Thank you for considering our rebuttal.

---

> > ### Comment · Reviewer_9sae · 2023-11-22
> > **An important remaining issue**
> >
> > Thank you for your response. Some of my concerts were answered, but we still do not agree on the most important issue.
> >
> > The issue with the partial analysis regarding backprop still remains.
> > You indeed wrote in (6) that backprop depends on the derivative WRT $\theta$, but then you did not analyze this derivative. As I explain in my review, this derivative may be a main contributor to the magnitude of the gradient in terms of $|x_i-x_j|$. Since your goal is to find sensitive terms WRT $|x_i-x_j|$, and you did not analyze a term that may be sensitive to $|x_i-x_j|$, the analysis is partial.

---

> ### Author Response · Authors · 2023-11-22
> **Further Response to Reviewer 9sae**
>
> Dear Reviewer 9sae,
>
> Thank you for your further feedback and we sincerely appreciate the opportunity to further address your concerns. Also, sorry for misunderstanding your question before. In what follows, please allow us to address your concerns on our partial analysis regarding backpropagation.
>
> We discuss immediately after Equation (6) the reason we focus on $\frac{\partial {\bf f}^{l+1}}{\partial {\bf f}^{l}}$. In particular, the gradient $\frac{\partial \mathcal{L}({\bf f}^{L})}{\partial \theta}$ propagates backward through the successive multiplication of $\frac{\partial {\bf f}^{l+1}}{\partial {\bf f}^{l}}$. As the layers progress deeper, the successive multiplication of $\frac{\partial {\bf f}^{l+1}}{\partial {\bf f}^{l}}$ becomes the predominant term on the overall stability of the gradient. In particular, the instability of $\frac{\partial {\bf f}^{l'+1}}{\partial {\bf f}^{l'}}$ may result in the instability of $\frac{\partial \mathcal{L}({\bf f}^{L})}{\partial \theta}$ for all parameters $\theta$ appearing at the $l$-th layer for any $l'\geq l$.
>
> Conversely, the other components, $\frac{\partial {\bf f}^{l}}{\partial \theta}$ and $\frac{\partial \mathcal{L}({\bf f}^{L})}{\partial {\bf f}^{L}}$, remain independent of the other layers. While $\frac{\partial \mathcal{L}({\bf f}^{L})}{\partial {\bf f}^{L}}$ appears in  $\frac{\partial \mathcal{L}({\bf f}^{L})}{\partial \theta}$ for all $\theta$, it is determined by the choice of loss function $\mathcal{L}$.
>
> In light of this, we propose Proposition 1 to delve into the details of $\frac{\partial {\bf f}^{l+1}}{\partial {\bf f}^{l}}$, showing that the terms $\frac{\partial {\bf x}_i^{l+1}}{\partial {\bf x}_j^{l}}$ -- for $l=0,\ldots,L-1$ that may appear in the derivative in Equation (6) -- are most sensitive to $||{\bf x}_i^l-{\bf x}_j^l ||$. This analysis motivates us to propose the regularization scheme in Equation (4).
>
>
>
> Thank you for considering our rebuttal. We are open to addressing any further questions or concerns on our paper.

---

### Official Review · Reviewer_wjdT · 2023-11-03

**Soundness:** 3 good
**Presentation:** 3 good
**Contribution:** 3 good
**Rating:** 6
**Confidence:** 3

**Summary:**

The paper highlights that there exists instability in training E(n)-graph neural networks (EGNN), especially maps for positions. The authors aim to stabilize the training of EGNN, especially by regularizations. Note that positional mapping in the EGNN includes the distance multiplication term, which is the distance between the node's position and its neighbor's, and it is critical to the E(n)-equivariance of EGNN.

The authors first point out that the parameters' gradient is proportional to the gradient of the EGNN outputs wrt the pairwise distance (between a node and its neighbor), and thus, the gradient term is the source of the instability of the EGNN training.

Due to the gradient term, the authors claim that the previous attempt to stabilize the training, i.e., the normalized distance multiplication, is not sufficient, and thus, the authors propose to penalize the norm of the gradient (of EGNN) wrt the pairwise distance.

To test the hypothesis, the papers compare the three versions of EGNNs (vanilla, normalized distance, and the proposed method) over various benchmark datasets.

**Strengths:**

I consider that the paper and its results are essential for several reasons:

1. The paper provides a better understanding of a critical problem in training EGNN, i.e., the instability of its training, especially to the potential audience unfamiliar with EGNN and other similar models.
2. The paper well motivates the proposed method so that readers can understand how each step contributes to the merits of the regularization.
3. I found that the paper has a well-organized structure that makes it clear to understand the proposed method.

In addition, I found that the paper has a well-organized structure that makes it clear to understand the proposed method.

**Weaknesses:**

While the proposed method seems well-motivated and interesting, the importance of the proposed method needs further analysis.

For example, it is unclear why the gradient wrt the pairwise distance is the key source of the instability of the EGNN training, while the authors claim that the instability stems from the gradient. However, the claim is backed only by the sensitivity analysis explained in Section 3.2, which seems close to intuitions in my understanding. I believe that it would be much clearer if the authors were showing that the gradient wrt the pairwise distance is exploded when the training of vanilla EGNN failed.

**Questions:**

In my understanding, when the gradients are unstable during training, one common solution is gradient norm penalty or gradient clipping. How does the proposed method perform against such common techniques?

---

> ### Author Response · Authors · 2023-11-15
> **Response to Reviewer wjdT**
>
> Thank you for your thoughtful review and valuable feedback. In what follows, we provide point-by-point responses to your comments.
>
>
> ---
>
> **Q1. It would be much clearer if the authors were showing that the gradient wrt the pairwise distance is exploded when the training of vanilla EGNN failed.**
>
>
> **Reply:** Thank you for your suggestion. We have included the plot of epochs vs. the gradient with respect to the pairwise distance, i.e. $\frac{\partial L}{\partial ||{\bf x}_i-{\bf x}_j||}$, in Appendix A.1. The numerical results indeed confirm that the gradient with respect to the pairwise distance explodes when the training of vanilla EGNN fails.
>
> ---
>
> **Q2. How does the proposed method perform against gradient norm penalty or gradient clipping?**
>
> **Reply:** By default, EDM has used gradient clipping in the experiment on QM9, and we have adopted gradient clipping for the baseline EDM, without regularization, for QM9 experiments. As shown in Section 5.2.2, regularization outperforms gradient penalty.
>
> As the reviewer suggested, we have further compared our proposed regularization against gradient norm penalty and gradient clipping in Appendix B.3. In particular, we consider training ENF and EDM with gradient clipping or gradient penalty -- using different clipping thresholds and gradient penalty hyperparameters -- on DW4 and compare them against our proposed regularization. The new results further confirm that our proposed regularization outperforms gradient penalty or gradient clipping by a remarkable margin.
>
> ---
>
> We have updated our submission based on the reviewer's feedback, with the revision highlighted in blue. We are happy to address further questions on our paper. Thank you for considering our rebuttal.

---

### Author Response · Authors · 2023-11-15
**General Response**

Dear reviewers and AC,

Thank you for your thoughtful reviews and valuable feedback, which have helped us significantly improve the paper. We appreciate the reviewers’ acknowledgment of the strengths of our paper. In this general response, we would like to first clarify our contribution and then summarize our revisions.

We fear reviewer 9sae ***misunderstood our proposed regularization scheme as simply adding factor 1 to the denominator of the coordinate update formula in EGCL***, i.e. Equation 1, which makes reviewer 9sae feel the contribution is incremental. We would like to stress that our proposed regularization is given by Equation 4 rather than adding factor 1 to the denominator of the coordinate update formula in EGCL. Adding factor 1 to the denominator of the coordinate update formula in EGCL is a default in ENF and EDM, which is not our contribution.


-----

Incorporating the comments and suggestions from all reviewers, besides fixing typos and restructuring and improving the presentation of the paper, we have made the following changes in the revision:

- Adding additional experimental results in Appendix A.1 to further confirm the instability issue of training ENF and EDM.

- Adding additional experimental results in Appendix B.3 to compare the performance of the proposed regularization against gradient clipping and gradient penalty.

-----

Thank you for considering our rebuttal.

---

### Author Response · Authors · 2023-11-20
**Seeking Feedback and Further Guidance**

Dear Reviewers,

As we approach the end of the discussion phase, we note that we have yet to receive any responses from your end. We fully appreciate that thoughtful reviewing requires time and effort, and we respect this aspect of the process.


However, we are eager to understand if there are any further concerns or areas for improvement in our submission that we can address. Your specialized insights and feedback are invaluable to the development of our work. If you could take the time to review our rebuttal and provide any additional comments, it would be greatly appreciated.


Thank you very much for your time and dedication to this process. We look forward to your valuable feedback.


Regards,


Authors

---

### Meta-Review · Area_Chair_oA3P · 2023-12-14

**Metareview:**

This paper studies instability for train equivariant normalizing flows (ENF), equivariant diffusion model (EDM). First, they identity the source of the instability by performing sensitivity analysis, and second they propose regularization method to resolve the problem. The proposals are justified by numerical experiments.

The paper is overall well written, and the authors pointed out an interesting issue. However, the main concern is that the investigation to identify the source of the instability is rather weak. Only some observations prove the existence of the numerical instability issue. The paper requires additional few rounds of revision before publication.

**Justification For Why Not Higher Score:**

As I have written in the metareview, the theoretical analysis in this paper is not thorough. What process causes the instability is remain unclear. Thus, I cannot recommend acceptance.

**Justification For Why Not Lower Score:**

N/A

---

### Decision · Program_Chairs · 2024-01-16

Reject